# Tox4 regulates transcriptional elongation and reinitiation during murine T cell development

Talang Wang[1,5], Ruoyu Zhao[1,2,5], Junhong Zhi [1,5], Ziling Liu [1,5], Aiwei Wu [1,5], Zimei Yang[1,3], Weixu Wang [4], Ting Ni [4], Lili Jing[3] & Ming Yu [1,2✉]

HMG protein Tox4 is a regulator of PP1 phosphatases with unknown function in development. Here we show that Tox4 conditional knockout in mice reduces thymic cellularity, partially blocks T cell development, and decreases ratio of CD8 to CD4 through decreasing proliferation and increasing apoptosis of CD8 cells. In addition, single-cell RNA-seq discovered that Tox4 loss also impairs proliferation of the fast-proliferating double positive (DP) blast population within DP cells in part due to downregulation of genes critical for proliferation, notably *Cdk1*. Moreover, genes with high and low expression level are more dependent on Tox4 than genes with medium expression level. Mechanistically, Tox4 may facilitate transcriptional reinitiation and restrict elongation in a dephosphorylation-dependent manner, a mechanism that is conserved between mouse and human. These results provide insights into the role of TOX4 in development and establish it as an evolutionarily conserved regulator of transcriptional elongation and reinitiation.

[1] Sheng Yushou Center of Cell Biology and Immunology, School of Life Sciences and Biotechnology, Shanghai Jiao Tong University, Shanghai 200240, China. [2] Department of Pathology, Shanghai Chest Hospital, Shanghai Jiao Tong University, Shanghai 200052, China. [3] School of Pharmacy, Shanghai Jiao Tong University, Shanghai 200240, China. [4] State Key Laboratory of Genetic Engineering, Collaborative Innovation Center of Genetics and Development, Human Phenome Institute, Shanghai Engineering Research Center of Industrial Microorganisms, School of Life Sciences and Huashan Hospital, Fudan University, Shanghai 200438, China. [5] These authors contributed equally: Talang Wang, Ruoyu Zhao, Junhong Zhi, Ziling Liu, Aiwei Wu. ✉email: mingyu@sjtu.edu.cn

Hematopoietic stem cells (HSCs) residing in adult bone marrow give rise to common myeloid progenitors (CMPs) and common lymphoid progenitors (CLPs), the latter of which are the common progenitors of two major types of lymphocytes in vertebrate adaptive immune system, T and B cells[1]. B cells complete their development in the bone marrow while precursors of T cells migrate to the thymus to complete their development. In the thymus, T cell precursors undergo β-, positive and negative selections to proliferate, and to achieve proper TCR expression and intermediate interaction strength between TCR and self-peptide-MHC complex[2,3].

CD4 and CD8 are commonly used makers for stages of αβ T cell development. Early T cell precursors in the thymus are called double negative (DN) cells for expressing neither CD4 nor CD8. The DN stage can be further divided into 4 stages, DN1-4, according to CD44 and CD25 expression[4,5]. β-selection occurs within the DN3 stage (CD44⁻CD25⁺), and ensures that only cells properly expressing a functional TCRβ proliferate and develop further into the CD4$^+$CD8$^+$ double positive (DP) stage[3]. Positive and negative selections occur likely independently rather than sequentially. Positive selection occurs in the cortex of the thymus, ensures successful rearrangement of the TCRα locus and rescues DP cells with TCRs exhibiting intermediate affinity towards self-peptide-MHC complex from apoptosis. Late in this process, DP cells give rise to single positive (SP) CD8 cytotoxic or CD4 helper T cells. It has been found recently that CD8$^+$ cells actually consist of two distinct populations, immature single positive (ISP), which are TCRβ⁻, and SP, which are TCRβ$^+$[6]. Negative selection occurs in both the cortex and the medulla of the thymus, and eliminates DP or SP cells with TCRs exhibiting high affinity towards self-peptide-MHC complex[2,7].

Once have completed their primary development in the thymus, T cells enter the bloodstream and then recirculate between blood and the secondary lymphatic tissues. Before encounter their specific antigens, they are called naive T cells and have condensed chromatin synthesizing little RNA or protein[8,9]. After encounter antigens, naive T cells become activated and differentiate into effector T cells. CD4 cells are capable of turning into several classes of effector T cells, including $T_H1$, $T_H2$, $T_H17$, $T_{FH}$, and regulatory T cells ($T_{reg}$)[10].

Transcription is the first step of gene expression, and cell type-specific transcription is fundamental to the development of multicellular organisms. Transcription is divided into three stages, initiation, elongation, and termination[11], and elongation in metazoans includes promoter escape, pause release, and productive elongation. Initiation and pause release are considered key check points of transcriptional regulation in metazoans[12,13]. Master transcription factors are known to play critical roles in metazoan development and cell differentiation. In T cell development and differentiation, over a dozen master regulators have been identified, including Bcl11b, Tox1 (usually called TOX), Zbtb7b (commonly known as ThPOK), etc[4,14]. Specifically, Bcl11b participates in T cell lineage commitment by down-regulating stem cell genes, inhibiting natural killer (NK) and myeloid gene expression, and maintaining T cell fate[15]; TOX is critical for CD4 cell development and the transcription of ThPOK[16,17]; ThPOK is critical for suppressing transcriptional program of CD8 cells[18,19].

Protein post-translational modifications, in particular phosphorylation, are known to play critical roles in transcription[20,21]. The C-terminal domain (CTD) of RNA polymerase II (Pol II) contains a heptad peptide (Y$^1$-S$^2$-P$^3$-T$^4$-S$^5$-P$^6$-S$^7$) that is repeated 26 and 52 times in budding yeast and human, respectively. The CTD plays critical and yet incompletely understood roles in gene expression and can be delicately regulated by dynamic phosphorylation of residues within this domain, most notably that of serine 2 (Ser-2) and serine 5 (Ser-5)[22,23]. The PP1 family of Ser/Thr phosphatases consists of PP1 α, β, and γ, which are ~90% identical in protein sequences, and is estimated to be responsible for dephosphorylation of around 50% of the human phosphoproteome[24]. They were recently found to be able to form the PTW protein phosphatase 1 (PTW/PP1) complex, which contains one of the phosphatases and three regulatory proteins, i.e., PNUTS, TOX4, and WDR82[25]. Among the PTW/PP1 complex regulatory subunits, PNUTS serves as a scaffold[25] and plays important roles in transcription and RNA processing by facilitating or suppressing dephosphorylation by PP1 phosphatases[26,27]; PNUTS and WDR82 were recently shown to prevent transcription-replication conflicts by promoting Pol II degradation[28]. We recently found that TOX4 is capable of directly binding PP1 phosphatases, restricts transcriptional elongation, and facilitates reinitiation[29]. TOX4 is a member of the TOX family transcription regulators, consisting of TOX1-4, which are capable of binding DNA and decompacting chromatin through their HMG box[30–34]. Among the TOX family members, Tox1 and 2 are well-known for being critical regulators of immune cell development and exhaustion of CD8 cells[31,35,36], Tox3 has been found to regulate transcription in neurons[37,38], whereas the role of TOX4 in development is unknown.

To understand the role of TOX4 in development, we performed mouse genetic and functional genomic studies. We found that Tox4 loss partially blocked DN and ISP to DP transition, impaired proliferation of both the DP blast population within DP and CD8, and increased apoptosis of CD8, and that Tox4 may be an evolutionarily conserved regulator of transcriptional elongation and reinitiation.

## Results

**Pan-hematopoietic Tox4 deletion reduces number of multipotential progenitors and impairs T cell development.** To understand the role of TOX4 in development, we generated *Tox4* conditional knockout mice by the CRISPR-Cas9 methodology, and two loxP sites in the same orientation were inserted upstream and downstream of exons 4–6, respectively (Supplementary Fig. 1a). Considering the critical roles of Tox1 and 2 in the immune system[16,30,31,33,34], we subsequently crossed mice with loxP-flanked *Tox4* alleles with transgenic mice expressing a Cre recombinase driven by the mouse *Vav1* promoter (*Vav1-Cre*) to generate mice with specific *Tox4* deletion in hematopoietic cells. The deletion of *Tox4* was efficient as determined by Western blot using whole cell lysate of thymocytes (Supplementary Fig. 1b). Flow cytometric analyses of bone marrow hematopoietic cells from 6- to 8-week-old *Tox4$^{f/f}$;Vav-Cre* conditional knockout (cKO) and littermate *Tox4$^{f/f}$* (control) mice discovered unaffected frequency of long-term HSCs (Lin⁻c-Kit$^+$Sca-1$^+$CD48⁻CD150$^+$) and short-term HSCs (Lin⁻c-Kit$^+$Sca-1$^+$CD48$^+$CD150$^+$)[39], while significant decreased frequency of the LSK (Lin⁻c-Kit$^+$Sca-1$^+$) population and multipotential progenitors (MPPs) (Lin⁻c-Kit$^+$Sca-1$^+$CD48$^+$CD150⁻) upon Tox4 loss (Supplementary Fig. 1c, d). In addition, frequency of erythroid (Supplementary Fig. 1e, f) and myeloid (Supplementary Fig. 1g, h) cells in bone marrow were unaffected. Moreover, bone marrow B cell development was minimally affected with some statistically significant small changes (Supplementary Fig. 1i, j), while frequency of lymphatic B cells was also unaffected (Supplementary Fig. 1k, l). Together, these results suggest that Tox4 may regulate the homeostasis of the MPP population.

Notably, we found that different from *Tox1* or *Tox2* knockout[16,31], *Tox4* knockout significantly reduces thymic cellularity (Supplementary Fig. 2a). Specifically, frequency and number of DP significantly decreased, frequency of CD4 significantly

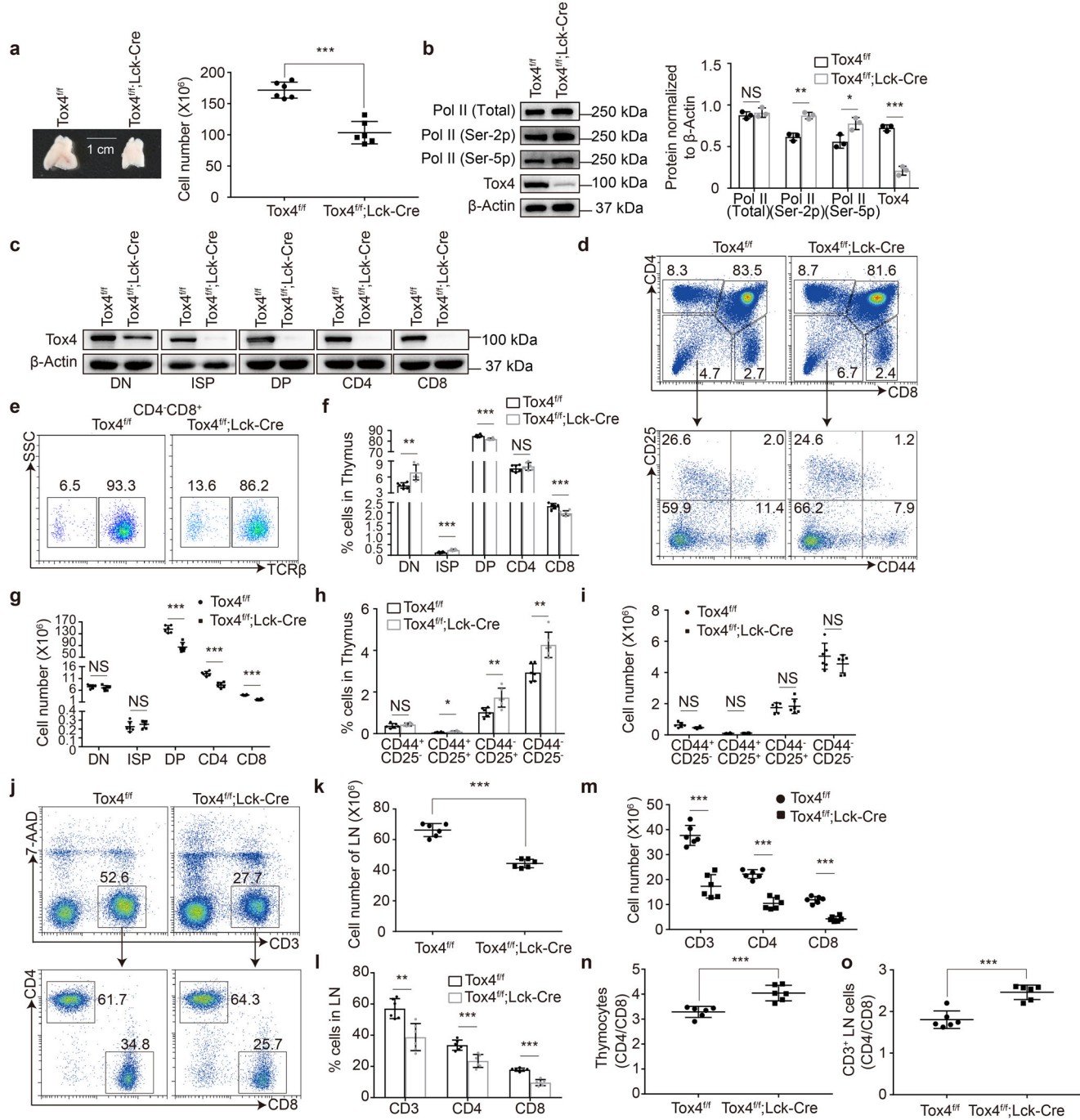

increased but its number was unaffected, frequency and numbers of DN and CD8$^+$ were unaffected (Supplementary Fig. 2b, c, e), and frequency and numbers of DN1 to 4 were also barely affected (Supplementary Fig. 2b, d, f). In addition, frequency of CD8 cells in the lymph nodes exhibited a significant decrease while that of CD3$^+$ or CD4 cells was unaffected (Supplementary Fig. 2g, h). Interestingly, the ratio of CD4 to CD8$^+$ cells significantly increased in both the thymus and the lymph nodes (Supplementary Fig. 2i, j). These results suggest Tox4 a regulator of T cell development.

**T cell-specific Tox4 knockout impairs T cell development in the thymus.** To further investigate the role of Tox4 in T cell development, we crossed *Tox4*$^{f/f}$ mice with transgenic mice containing a Cre-recombinase gene driven by the proximal

promoter of the lymphocyte-specific protein tyrosine kinase (*Lck*) gene (*Lck-Cre*) to generate mice with T cell-specific deletion of *Tox4* starting from the DN3 stage. Tox4 knockout significantly reduces thymic cellularity (Fig. 1a), which is similar to what we have found with pan-hematopoietic *Tox4* deletion mice (Supplementary Fig. 2a), and the deletion was efficient as determined by Western blot using whole cell lysate of unsorted and sorted thymocytes from *Tox4*$^{f/f}$;*Lck-Cre* (conditional knockout, cKO) and littermate *Tox4*$^{f/f}$ (control) mice (Fig. 1b, c). Tox4 loss also increases the level of Ser-5 phosphorylated (Ser-5p) and Ser-2 phosphorylated (Ser-2p) Pol II (Fig. 1b), which is consistent with what we have found in K562 cells[29]. Flow cytometric comparison of thymocytes from cKO and control mice uncovered significantly decreased frequency and number of DP, significantly increased frequency but unaffected numbers of DN and ISP,

**Fig. 1 Tox4 conditional knockout in mice impairs T cell development. a** Tox4 KO reduces thymic cellularity. Left: thymi image of *Tox4^f/f* (control) and *Tox4^f/f;Lck-cre* (cKO) mice showing size difference. A scale bar was used in the image. Right: a scatter plot comparing thymic cellularity of control and cKO mice. **b** Western blot comparing cellular level of Tox4, total, Ser-5 phosphorylated, and Ser-2 phosphorylated Pol II in control and Tox4 cKO thymocytes. β-Actin was used as a loading control. Left: representative pictures of Western blot; Right: a bar graph comparing relative level of total Pol II, Pol II (Ser-2p), Pol II (Ser-5p), and Tox4 in control and cKO cells quantified by ImageJ. Pictures are representative of three independent experiments ($n = 3$). Statistical significance was determined with a two-sided Student's t-test; the centers and the error bars represent the mean and the SD, respectively. NS: $P \geq 0.05$, *$P < 0.05$, **$P < 0.01$, and ***$P < 0.001$. **c** Western blot comparing cellular level of Tox4 in sorted thymocytes from control and Tox4 cKO mice. **d** Representative plots of flow cytometric analysis of expression of CD4, CD8, CD25, and CD44 in thymocytes. **e** Representative plots of flow cytometric analysis of expression of TCRβ in the CD4−CD8+ population. **f, h** Bar graphs comparing frequency of DN, DP, and SP cells (**f**) and DN1-4 cells (**h**) in control and cKO mice. DN: CD4−CD8−, ISP: TCRβ−CD4−CD8+, DP: CD4+CD8+, CD4 SP: CD4+CD8−, CD8 SP: TCRβ+CD4−CD8+, DN1: CD4−CD8−CD44+CD25−, DN2: CD4−CD8−CD44+CD25+, DN3: CD4−CD8−CD44−CD25+, DN4: CD4−CD8−CD44−CD25−. **g, i** Scatter plots comparing numbers of DN, ISP, DP, and SP cells (**g**) and DN1-4 cells (**i**) in control and cKO mice. **j** Representative plots of flow cytometric analysis of expression of CD3, CD4, and CD8 in lymphocytes. **k** A scatter plot comparing numbers of lymphocytes of control and cKO mice. **l** A bar graph comparing frequency of T cell subpopulations within lymphocytes of control and cKO mice. **m** A scatter plot comparing numbers of CD3+, CD8, and CD4 cells in the lymph nodes in control and cKO mice. **n, o** Scatter plots comparing ratio of CD4 to CD8 cells in the thymus (**n**) and the lymph nodes (**o**) in control and cKO mice. Pictures in **d**, **e** and **j** are representative of six independent experiments ($n = 6$). Statistical significance was determined with a two-sided Student's t-test; the centers and the error bars represent the mean and the SD, respectively. NS: $P \geq 0.05$, *$P < 0.05$, **$P < 0.01$, and ***$P < 0.001$.

unaffected frequency but decreased numbers of CD4, and decreased frequency and number of CD8 (Fig. 1d–g), mostly consistent with what we have found using mice with pan-hematopoietic *Tox4* deletion (Supplementary Fig. 2b, c, e). In addition, frequency and number of DN1 were unaffected; frequency of DN2-4 increased while their numbers were unaffected (Fig. 1d, h, i). Together, these results suggest partially blocked T cell development from DN and ISP to DP stage under Tox4 deficiency condition, which likely contributes to the thymic cellularity reduction.

Flow cytometric comparison of lymphocytes from cKO and control mice discovered significantly reduced total cell number and frequency and numbers of CD3+, CD8, and CD4 (Fig. 1j–m). Moreover, ratio of CD4 to CD8 cells significantly increased in both the thymus and the lymph nodes after *Tox4* deletion (Fig. 1n, o), which is consistent with what we have found with pan-hematopoietic *Tox4* deletion mice (Supplementary Fig. 2i, j). Together, these results suggest that the development of CD8 cells are more dependent on Tox4 than that of CD4 cells.

**Tox4 deficiency impairs proliferation and increases apoptosis of CD8 cells.** To further investigate causes for the number reduction of T cells in the thymus and the lymph nodes, we performed proliferation and apoptosis analyses. Ki67 staining discovered slightly accelerated proliferation of DN, ISP, and DP thymocytes (Fig. 2a, b), unaffected proliferation of CD4 thymocytes, slightly decelerated proliferation of CD8 thymocytes (Fig. 2a, b), accelerated proliferation of lymphatic CD3+ and CD4 cells, and minimally affected proliferation of lymphatic CD8 cells (Fig. 2c, d) after *Tox4* deletion. These results suggest that proliferation defect may contribute to the frequency and number reduction of CD8 cells (Fig. 1f, g, l, m), resulting in the decrease of CD8 to CD4 ratio in both the thymus and the lymph nodes (Fig. 1n, o). Moreover, 7-AAD and Annexin V staining discovered significantly increased frequency of early apoptotic (7-AAD−Annexin V+) CD8 thymocytes, slightly increased frequency of early apoptotic DP cell, unaffected frequency of early apoptotic DN, ISP, and CD4 thymocytes (Fig. 2e, f), and significantly increased frequency of early apoptotic lymphatic T cells after *Tox4* deletion (Fig. 2h, i). With respect to frequency of late apoptotic (7-AAD+Annexin V+) cells, only CD8 cells in both the thymus and the lymph nodes showed significant increase, while the rest of the populations showed no change (Fig. 2e, g, h, j). These results suggest that increased apoptosis of CD8 also contributes to the frequency and number reduction of CD8 cells

(Fig. 1f, g, l, m) in addition to impaired proliferation of CD8 thymocytes (Fig. 2a, b).

**Tox4 deficiency impairs activation and proliferation of CD8 cells.** To determine if Tox4 is required for antigen-stimulated T cell responses, we isolated naive CD4 and CD8 cells from control and cKO mice, respectively, and stimulated them with CD3 and CD28 antibodies. Activated T cells upregulate CD69 and CD25[40]. We found that CD4 cells from control and cKO mice upregulated CD69 and CD25 to comparable level 12 or 36 h after stimulation (Fig. 3a–d). In addition, CFSE assays showed unaffected proliferation of CD4 cells upon *Tox4* deletion (Fig. 3e). Together, these results suggest that Tox4 does not regulate activation and proliferation of CD4 cells. In contrast, CD8 cells from cKO mice were unable to upregulate CD69 and CD25 as efficiently as those from control mice 12 h after stimulation, but were able to achieve expression level of them comparable to that of control cells 36 h after stimulation (Fig. 3f–i). Moreover, CFSE assays showed slightly impaired proliferation of CD8 cells (Fig. 3j). Together, these results suggest that Tox4 plays an accessory role in activation and proliferation of CD8 cells.

**Tox4 deficiency impairs proliferation of the DP blast population within DP cells.** To further understand the role of Tox4 in T cell development, we compared thymocytes of control and cKO mice by single-cell RNA-seq (scRNA-seq), and two pairs of control and cKO mice were analyzed. To identify subpopulations within thymocytes, we performed graph-based clustering and found almost all the major populations known to exist in the thymus, including αβ T cells (15 clusters, 95.22%), γδ T cells (1 cluster, 0.21%), B cells (1 cluster, 0.33%), mTEC (1 cluster, 0.38%), and the mononuclear phagocyte system (MPS) (2 clusters, 3.87%) (Fig. 4a). The 15 clusters of αβ T cells were further grouped into 6 developmental stages as previously described[41], including DN, DP blast (DPbla), DP undergoing rearrangement (DPre), DP cells under selection (DPsel), CD4 SP thymocytes (CD4SP) and CD8 SP thymocytes (CD8SP) (Fig. 4a). Many recognized markers of T cell development were found to be specifically expressed in the corresponding clusters or stages (Supplementary Fig. 3a, b). To validate the clustering and grouping results, we performed trajectory analysis of αβ T cells using Monocle3, and found that the inferred trajectory is consistent with the known order of T cell development (Fig. 4b, c). Together, these results suggest that the clustering and grouping results are valid.

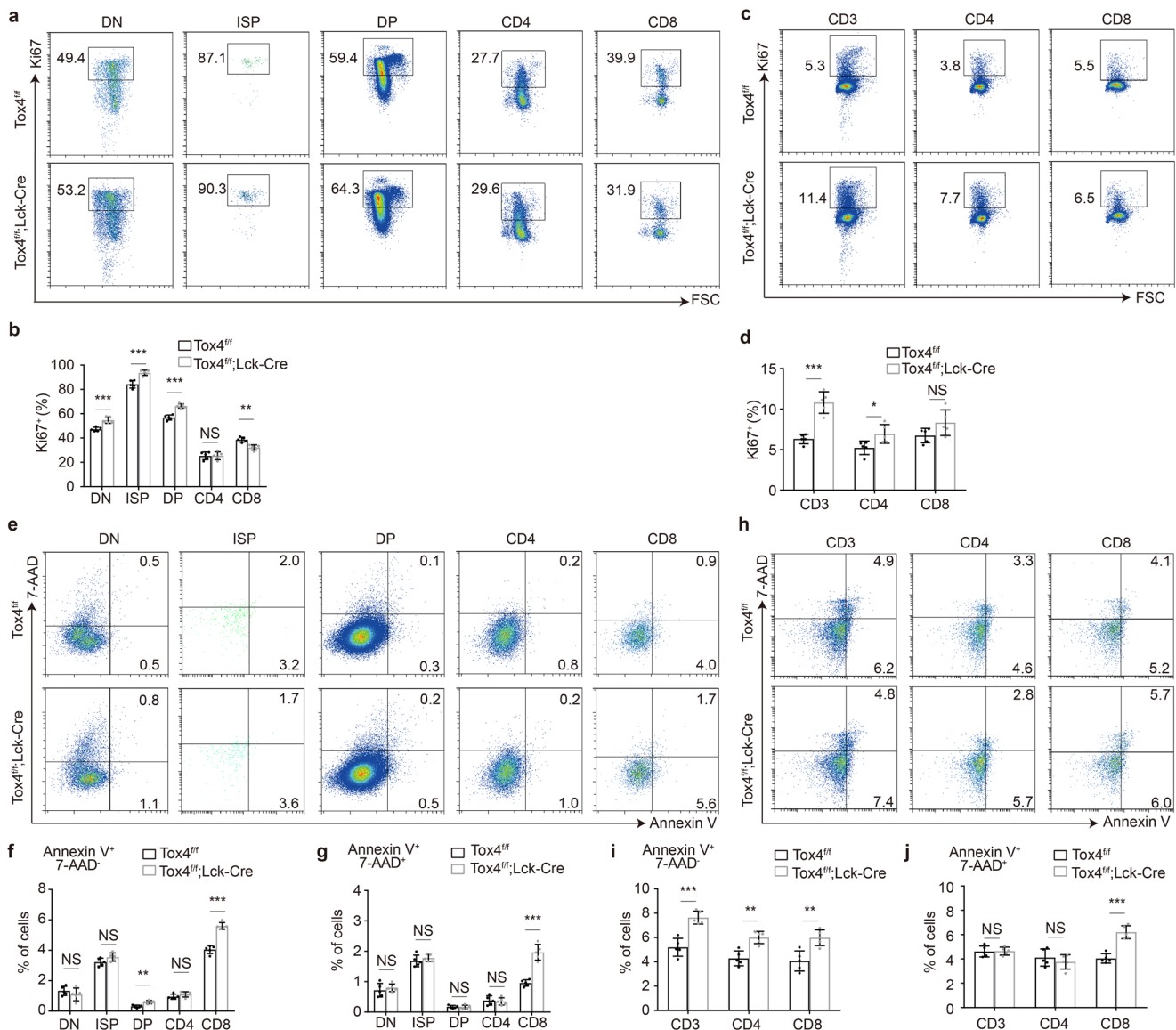

**Fig. 2 Tox4 loss impairs proliferation and increases apoptosis of CD8 cells. a**, **c** Representative plots of flow cytometric analyses of Ki67 expression for cell proliferation in thymocytes (**a**) and lymphatic T cells (**c**). **b**, **d** Bar graphs comparing frequency of Ki67 positive cells of T cell subpopulations in the thymus (**b**) and the lymph nodes (**d**) in control and cKO mice. **e**, **h** Representative plots of flow cytometric analyses of 7-AAD and Annexin V staining for apoptosis in thymocyte (**e**) and lymphatic T cells (**h**). **f**, **i** Bar graphs comparing frequency of early apoptotic (Annexin V$^+$7-AAD$^-$) cells of T cell subpopulations in the thymus (**f**) and the lymph nodes (**i**) in control and cKO mice. **g**, **j** Bar graphs comparing frequency of late apoptotic (Annexin V$^+$7-AAD$^+$) cells of T cell subpopulations in the thymus (**g**) and the lymph nodes (**j**) in control and cKO mice. Pictures in **a**, **c**, **e** and **h** are representative of five independent experiments ($n = 5$). Statistical significance was determined with a two-sided Student's t-test; the centers and the error bars represent the mean and the SD, respectively. NS: $P \geq 0.05$, *$P < 0.05$, **$P < 0.01$, and ***$P < 0.001$.

To identify the stages of αβ T cell development that were affected by Tox4 loss, we calculated relative cell count for each cluster to the unchanged DN cells (Fig. 1g) for control and cKO thymocytes, respectively. We found that starting from DPbla2, the relative cell count of each αβ T cell cluster became smaller in cKO mice relative to that in control mice except for that of DPre7, suggesting that Tox4 also regulates homeostasis of DP cells (Fig. 4d) in addition to DN and ISP to DP transition (Fig. 1d–i) during T cell development. To determine if Tox4 loss affects proliferation of those clusters, we calculated proliferation score by averaging expression of proliferation-related genes for cells in each cluster. We found that DPbla2 had not only the highest but also the most significantly decreased proliferation score upon *Tox4* deletion (Fig. 4e and Supplementary Fig. 4a). In addition, DPbla1 not only had the second highest proliferation

score but also was the other cluster besides DPbla2 exhibiting significant proliferation score decrease (Fig. 4e). Together, these results suggest that decreased proliferation of DPbla1 and 2 may contribute to the cell number reduction of DP upon Tox4 loss. Moreover, exclusive of DN, DPbla1, and DPbla2, 6 of the remaining 12 clusters of αβ T cells, i.e., DPre2-6 and DPsel1, exhibited increased proliferation scores (Fig. 4e), which are consistent with the flow cytometric results showing increased proliferation of DP cells upon Tox4 loss when analyzed in bulk (Fig. 2a, b). Furthermore, proliferation scores of CD4 thymocytes were unaffected by *Tox4* deletion (Fig. 4e), which is also consistent with the flow cytometric results (Fig. 2a, b).

It was found previously that frequency of actively cycling (non-G1) cells of highly proliferating population is higher than that of low proliferating population[42]. To assess if Tox4 loss affects the

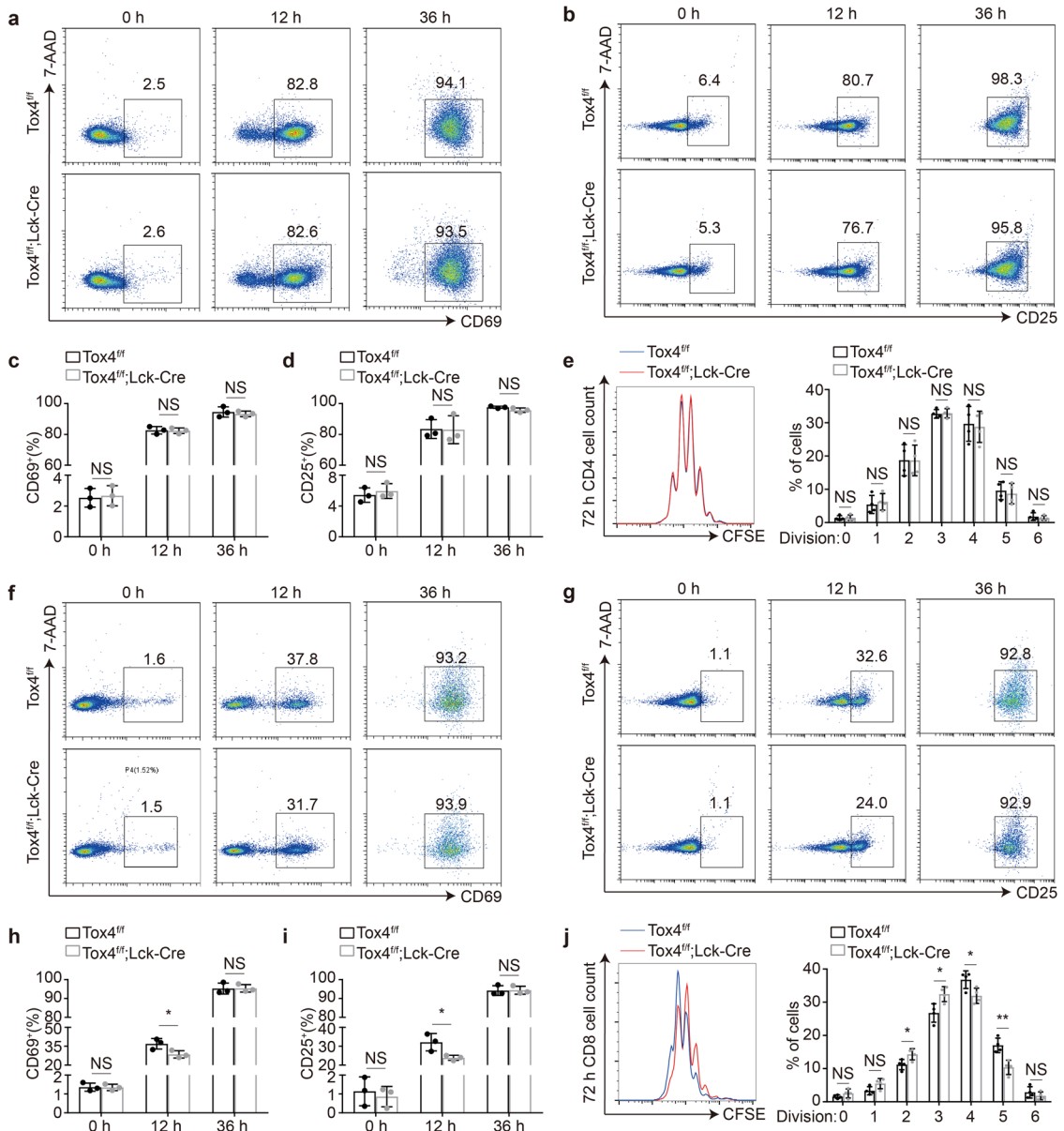

**Fig. 3 Tox4 deficiency impairs activation and proliferation of CD8 cells. a**, **b** Representative plots of flow cytometric analyses of CD69 (**a**) and CD25 (**b**) expression of stimulated CD4 lymphocytes from control and cKO mice. **c**, **d** Bar graphs comparing frequency of CD69+ (**c**) and CD25+ (**d**) CD4 lymphocytes from control and cKO mice. **e** Analysis of proliferation of activated CD4 by CFSE staining. Left: Representative plots of flow cytometric analyses of CFSE-labeled CD4 lymphocytes from control and cKO mice. Right: A bar graph comparing frequency of CFSE-labeled CD4 lymphocytes from control and cKO mice. **f**, **g** Representative plots of flow cytometric analyses of CD69 (**f**) and CD25 (**g**) expression of stimulated CD8 lymphocytes from control and cKO mice. **h**, **i** Bar graphs comparing frequency of CD69+ (**h**) and CD25+ (**i**) CD8 lymphocytes from control and cKO mice. **j** Analysis of proliferation of activated CD8 by CFSE staining. Left: Representative plots of flow cytometric analyses of CFSE-labeled CD8 lymphocytes from control and cKO mice. Right: A bar graph comparing frequency of CFSE-labeled CD8 lymphocytes from control and cKO mice. Pictures in **a**, **b**, **f** and **g** are representative of three independent experiments ($n = 3$). Pictures in **e** and **j** are representative of four independent experiments ($n = 4$). Statistical significance was determined with a two-sided Student's t-test; the centers and the error bars represent the mean and the SD, respectively. NS: $P \geq 0.05$, *$P < 0.05$, and **$P < 0.01$.

frequency of cycling cells, we quantified cells of G1, S, and G2/M phases of the cell cycle for each cluster according to the expression of cell cycle-related genes, and found that the frequency changes of actively cycling cells of the highly proliferating clusters, DPbla1 and 2, are no greater than those of the rest of the clusters (Supplementary Fig. 4b, c). In addition, it has been found recently by scRNA-seq that a subset of G2 phase cells exit to G0 phase after completion of the ongoing cycle due to low mitogen signal and/or low CDK activity[42], which raises a possibility that gene expression changes of clusters with

high proliferation scores upon Tox4 loss may induce cell cycle exit. We therefore analyzed gene expression changes for the 15 αβ T cell clusters (Fig. 4f–h and Supplementary Fig. 4d). For the 3 clusters with the highest proliferation scores, i.e., DPbla1, DPbla2, and DPre1, 15, 66 and 24 differentially expressed genes were identified, respectively. Notably, *Cdk1*, *Ccnb1*, *Ccnb2*, and *Ccna2* were downregulated in DPbla2 (Fig. 4g, i). The results can be validated by qRT-PCR using DP blast cells (CD4+CD8+FSChiCD69lo) sorted from control and cKO mice, respectively (Supplementary Fig. 5a–g). To obtain enough cells

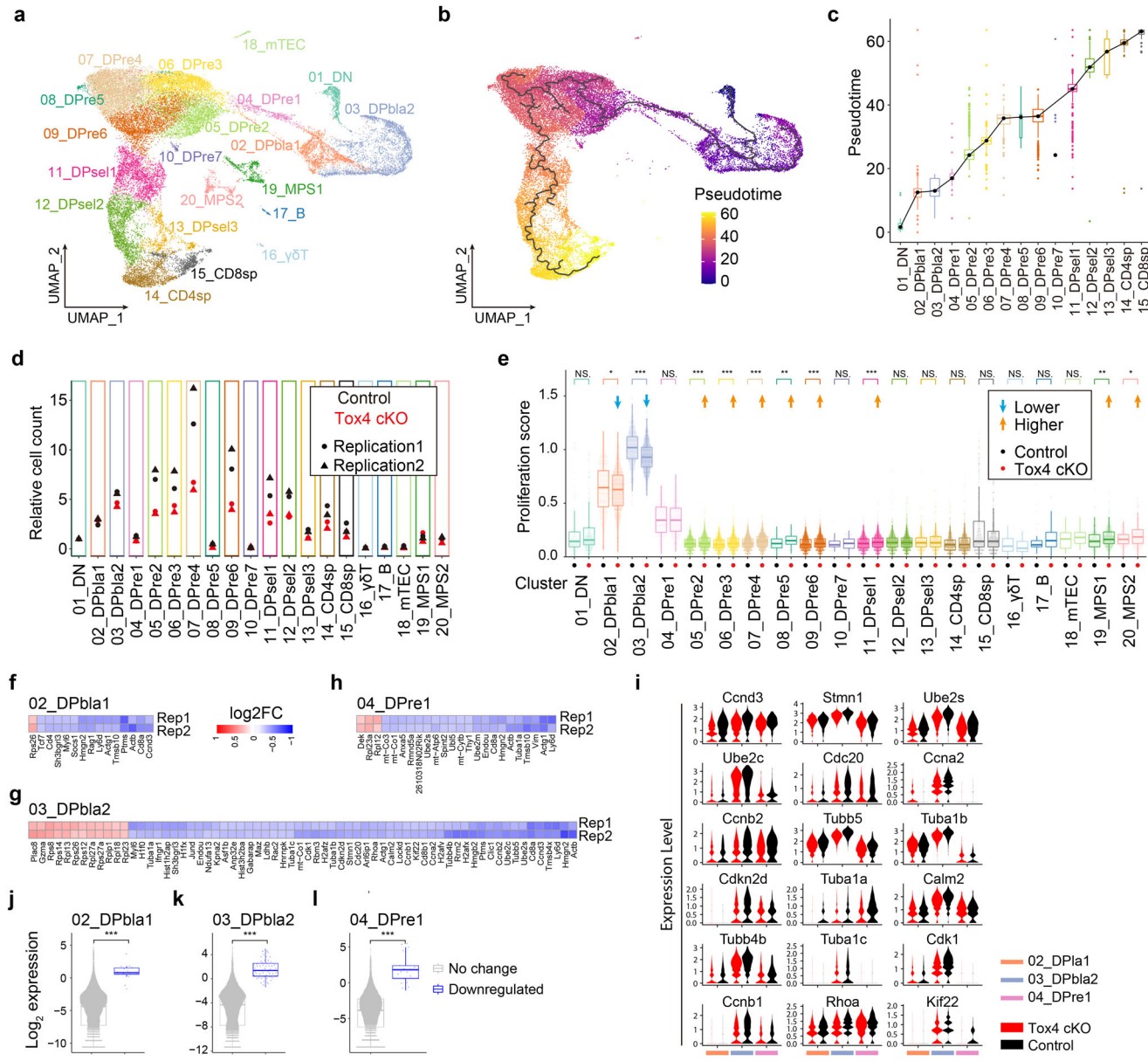

**Fig. 4 Tox4 loss impairs proliferation of the DP blast population within DP cells. a** Combined UMAP plot showing clusters of all cells from thymi of 2 pairs of control and cKO mice. **b** Two-dimensional representation of αβ T cells via UMAP, as colored by the Monocle3 estimated pseudotime. Each dot represents one cell, and black lines represent Monocle3 estimated developmental trajectory. **c** Boxplot of Monocle3 predicted pseudotime of αβ T cells colored by cluster. Black dots represent median pseudotime values of cell clusters, and they are connected with black lines to manifest the increasing median pseudotime of cell clusters along the estimated developmental trajectory. A small cluster, 10_DPre, was skipped for straying far away from the trendline. The width of each box is proportional to the size of that cluster. **d** Cell count of each cluster relative to that of DN cells (cluster 01_DN). The relative cell count of DN cells for each biological replicate is set as 1. **e** Boxplots with beeswarm plots of proliferation scores calculated by averaging expression of a set of proliferation signature genes for each cluster in control and Tox4 cKO mice. **f–h** Heatmaps showing log2 fold change of differentially expressed genes of 02_DPbla1 (**f**), 03_DPbla2 (**g**), and 04_DPre1 (**h**) clusters in Tox4 cKO thymocytes relative to control thymocytes. **i** Violin plots showing expression of cell-cycle-related differentially expressed genes of 02_DPbla1, 03_DPbla2, and 04_DPre1 clusters. **j–l** Violin plots comparing expression level of downregulated and unaffected genes upon Tox4 loss within 02_DPbla1, 03_DPbla2, and 04_DPre1 clusters. For boxplots in (**c**, **e**, and **j–l**), the standard boxplot notation was used (lower/upper hinges–first/third quartiles; whiskers extend from the hinges to the largest/smallest values no further than 1.5 x inter-quartile ranges; middle line–the median). Data beyond the end of the whiskers are called "outlying" points and are plotted individually in (**c**). Two independent experiments were performed (*n* = 2). The differences of proliferation score between Tox4 cKO and control cells (**e**) and the differences of expression between two groups of genes (**j–l**) were tested using two-sided Wilcoxon rank-sum test. NS: *P* ≥ 0.05, \**P* < 0.05, \*\**P* < 0.01, and \*\*\**P* < 0.001.

for qRT-PCR, a less stringent threshold for selecting CD69⁻ cells were used (Supplementary Fig. 5a). The changes would be more significant if a more stringent threshold was used (Supplementary Fig. 5b–g). Moreover, there is a positive correlation between G2M score calculated by expression of makers of G2 and M phases and

the expression of Cdk1 (Supplementary Fig. 5h). Together, these results suggest that low Cdk1 activity may contribute to the proliferation defect of DPbla2 and lead to cell cycle exit.

We also found that within each cluster, downregulated genes are mainly those with high expression level (Fig. 4j–l and

Supplementary Fig. 4e), suggesting that genes with high expression level are more sensitive to Tox4 deficiency than those with medium or low expression level. In addition, we found that *Cd8a* and *b1* were downregulated in several clusters (Fig. 4f–h and Supplementary Fig. 4d), the results can also be validated by qRT-PCR using DP blast cells sorted from control and cKO mice, respectively (Supplementary Fig. 5a–c), but protein level of Cd8 was minimally affected (Fig. 1d, j), suggesting it unlikely to cause the increase of CD4 to CD8 ratio (Fig. 1n, o).

**Tox4 loss affects transcription of a small subset of genes in DP cells.** To facilitate the mechanistic analyses, we started by identifying direct target genes of Tox4. By CUT&Tag using DP cells purified from control and cKO mice, we identified 11,966 Tox4 peaks (Fig. 5a, Supplementary Fig. 6a, b), and found that Tox4 occupancy pattern resembles that of Pol II (Figs. 5b and 6a), which is similar to what we have found with TOX4 in K562 cells[29]. By RNA-seq using RNA of DP cells sorted from control and cKO mice, respectively, we found that the numbers of downregulated and upregulated genes were 68 and 91, respectively (Fig. 5c). Comparative analyses of the CUT&Tag and the RNA-seq data identified 82 Tox4 direct target genes with 29 downregulated and 53 upregulated with fold change ≥ 1.5 and FDR < 0.05 (Fig. 5d).

The numbers of differentially expressed genes and Tox4 direct target genes are small, we therefore analyzed effects of *Tox4* deletion on nascent RNA synthesis by performing TTchem-seq experiments[43,44]. We found that Tox4 loss only affected transcriptional output of several hundred transcripts (Fig. 5e±g). With fold change ≥ 1.2 and FDR < 0.05, the numbers of genes with decreased and increased output were 182 and 249, respectively (Fig. 5f), and among them, the numbers of Tox4 direct target genes with decreased and increased output were 127 and 153, respectively (Fig. 5g). Moreover, comparative analysis direct target genes identified by CUT&Tag and RNA-seq and direct target genes identified by CUT&Tag and TTchem-seq obtained an overlap of 30 genes with 9 downregulated in both groups, 19 upregulated in both groups and 2 downregulated in RNA-seq but exhibiting increased output in TTchem-seq (Fig. 5h). Together, these results suggest that Tox4 affects transcription of a small subset of genes in DP cells. Considering that DPbla, DPre, and DPsel make up 17.8%, 60.0%, and 22.2% of total DP cells, and that 12 clusters of DP cells showed differences in differentially expressed genes upon Tox4 loss (Fig. 4f–h and Supplementary Fig. 4d), one of the reasons why *Tox4* deletion only significantly affected expression of a small subset of genes may be the heterogeneity of the total DP population.

Gene Ontology (GO) analysis of upregulated and downregulated Tox4 direct target genes identified by CUT&Tag and RNA-seq (Fig. 5d) was unable to identify any significantly enriched GO terms due to the small numbers; in contrast, GO analyses of upregulated and downregulated Tox4 direct target genes identified by CUT&Tag and TTchem-seq (Fig. 5g) were able to identify some significantly enriched GO terms, including lymphocyte differentiation, T cell differentiation, regulation of hematopoiesis, etc (Fig. 5i, j). Notably, downregulated, upregulated and unaffected genes tend to have high, low-to-medium, and medium expression level, respectively (Fig. 5k, l), and further analyses of our published data of K562 cells[29] reached very similar conclusions (Fig. 5m, n). These results are also partially in agreement with the scRNA-seq results (Fig. 4j–l and Supplementary Fig. 4e) showing that highly expressed genes are dependent on Tox4 (Fig. 4j–l). The reason why scRNA-seq did not show that upregulated genes were mainly those with low expression level is that the low sensitivity of scRNA-seq technology makes low-expression genes hard to be detected.

**Tox4 loss affects transcription of a small subset of genes in CD8 thymocytes.** To further understand how Tox4 deficiency decreases ratio of CD8 to CD4 cells (Fig. 1n, o) through impairing proliferation of CD8 thymocytes (Fig. 2a, b) and increasing apoptosis of CD8 cells (Fig. 2e–j), we sorted CD8 thymocytes (TCRβ⁺CD8⁺) from control and cKO mice, respectively, and performed RNA-seq experiments. Tox4 deficiency only significantly affected mRNA level of a small subset of genes, and with fold change ≥ 1.5 and FDR < 0.05, the numbers of downregulated and upregulated genes were 10 and 34, respectively (Supplementary Fig. 7a, b). Although no GO term was found to be enriched through GO analysis, *Cdkn1a* (encoding p21), a well-known negative regulator of the cell cycle and a direct target of Tox4 in DP cells (Fig. 5d), was significantly upregulated under Tox4 deficiency, suggesting that p21 upregulation may contribute to the impaired proliferation of CD8 thymocytes. In addition, expression of several apoptosis-related genes was significantly affected, including *Ntrk3*[45,46], *Gimap7*[47], and *Nfia*[48], under Tox4 deficiency, suggesting that dysregulation of these genes may contribute to the increased apoptosis of CD8 cells.

**Tox4 regulates expression of a small subset of extragenic transcripts in DP cells.** It is known that WDR82, PNUTS, PP1, and SET1 restrict extragenic transcription[49], and we found recently that TOX4 regulates extragenic transcription in K562 cells and may mainly play a restrictive role[29]. To determine if it is also the case in DP cells, we performed comparative analyses using Tox4 CUT&Tag data with the RNA-seq data or the TTchem-seq data. For the analysis using CUT&Tag and TTchem-seq, the numbers of extragenic transcripts with decreased and increased output were 872 and 974, respectively, with fold change > 1.5, RPKM ≥ 0.1 and FDR < 0.01, and among them, the numbers of Tox4 direct targets with decreased and increased output were 177 and 189, respectively (Supplementary Fig. 8a). For the analysis using CUT&Tag and RNA-seq, the numbers of extragenic transcripts with decreased and increased expression were 1072 and 1599, respectively, with fold change > 1.5, RPKM ≥ 0.1 and FDR < 0.01, and among them, the numbers of Tox4 direct targets with decreased and increased expression were 58 and 540, respectively (Supplementary Fig. 8b). Snapshots of two Tox4 regulated extragenic transcripts identified by CUT&Tag and TTchem-seq and snapshots of two Tox4 regulated extragenic transcripts identified by CUT&Tag and RNA-seq are shown (Supplementary Fig. 8c, d). With respect to why TTchem-seq identified a smaller number of differentially expressed extragenic transcripts compared to that identified by RNA-seq, one reason may be that the majority of the DP cells, i.e., DPre and DPsel, are likely to be transcriptionally inactive just like native T cells in the secondary lymphatic tissues[8,9], and another reason may be the heterogeneity of DP cells. Annotation of Tox4 regulated extragenic transcripts identified by CUT&Tag and TTchem-seq discovered that it mainly facilitates transcription of PROMPT, and restricts transcription of lncRNA and transcriptional readthrough (Supplementary Fig. 8a). Annotation of Tox4 regulated extragenic transcripts identified by CUT&Tag and RNA-seq obtained results different from the results above, but they may not be as reliable as the results above for the reason that RNA-seq measure stable RNA while TTchem-seq measure nascent RNA.

**Tox4 may restrict elongation and facilitate reinitiation in DP cells.** To understand mechanisms underlying Tox4 mediated transcription in murine T cells, we first examined the effects of *Tox4* deletion on Pol II occupancy by CUT&Tag experiments for total, Ser-5 phosphorylated and Ser-2 phosphorylated Pol II. Correlation analyses of related biological replicates suggest that the data are highly reproducible (Supplementary Fig. 9a–c). We

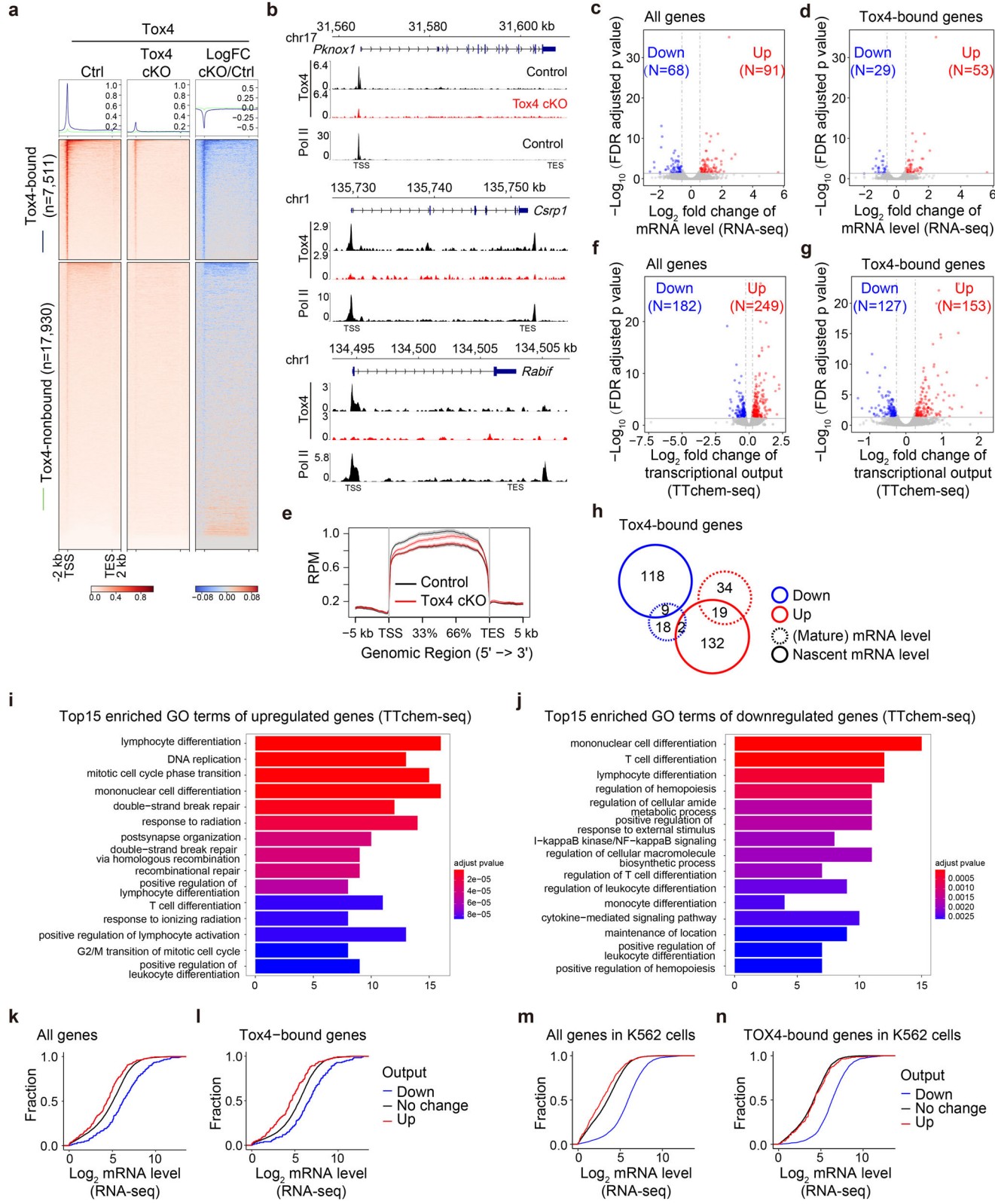

found that Tox4 loss decreased total Pol II occupancy (Fig. 6a, j and Supplementary Fig. 9d), which is consistent with what we have found in K562 cells[29]. The numbers of Tox4-bound genes exhibiting significantly decreased and increased Pol II occupancy near transcription start sites (TSSs) were 84 and 5, respectively (Fig. 6b), and the corresponding numbers near transcription end sites (TESs) were 6 each (Fig. 6c). In addition, Tox4 loss increased

Pol II (Ser-2p) occupancy (Fig. 6d, j and Supplementary Fig. 9e), which is consistent with the Western blot results (Fig. 1b) but slightly different from what we have found in K562 cells showing slightly decreased occupancy of it[29]. The numbers of Tox4-bound genes exhibiting significantly decreased and increased Pol II (Ser-2p) occupancy near TSSs were 93 and 537, respectively (Fig. 6e), and the corresponding numbers near TESs were 87 and 1471,

**Fig. 5 Tox4 loss affects expression of a small subset of genes in DP cells. a** Genome-wide meta-gene profiles and heatmaps of CUT&Tag comparing chromatin occupancy of Tox4 in Tox4 cKO versus control (Ctrl) cells. Genes with and without significant Tox4 binding are presented separately. **b** Normalized read distribution of CUT&Tag of Tox4 and Pol II within the *Pknox1, Csrp1, Rabif* loci in Tox4 cKO versus control cells. **c, d** Volcano plots showing mRNA level changes of genes (**c**) and Tox4-bound genes (**d**) in cKO versus control cells. **e** Meta-gene profiles of TTchem-seq of protein-coding genes in cKO versus control cells. **f, g** Volcano plots showing transcriptional output changes of genes (**f**) and Tox4-bound genes (**g**) in Tox4 cKO versus control cells. **h** A Venn diagram showing overlaps between Tox4 direct targets identify by CUT&Tag and RNA-seq and direct targets identified by CUT&Tag and TTchem-seq. **i, j** GO analysis results of upregulated (**i**) and downregulated (**j**) genes upon *Tox4* deletion. **k, l** Cumulative frequency curves comparing mRNA level of upregulated, unaffected, and downregulated genes (**k**) and Tox4 direct target genes (**l**) upon *Tox4* deletion in murine DP cells. **m, n** Cumulative frequency curves comparing mRNA level of upregulated, unaffected, and downregulated genes (**m**) and TOX4 direct target genes (**n**) upon TOX4 loss in K562 cells.

respectively (Fig. 6f). Moreover, Tox4 loss increased Pol II (Ser-5p) occupancy (Fig. 6g, j and Supplementary Fig. 9f), which is also consistent with the Western blot results (Fig. 1b) but slightly different from what we have found in K562 cells showing slightly decreased occupancy of it[29]. The numbers of Tox4-bound genes exhibiting significantly decreased and increased Pol II (Ser-5p) occupancy near TSSs were 0 and 71, respectively (Fig. 6h), and the corresponding numbers near TESs were 0 and 33, respectively (Fig. 6i). Normalization of Ser-5 and Ser-2 phosphorylated Pol II occupancy individually by total Pol II occupancy discovered increased relative occupancy (Fig. 6d, g, Supplementary Fig. 9e, f), which is consistent with what we have found in K562 cells[29]. Together, these results suggest that Tox4 may restrict elongation in murine DP cells. To directly measure the effect of Tox4 deficiency on elongation rate, we performed 4sUDRB-seq[29] using DP cells. We were unable to obtain enough RNA for sequencing library construction, again supporting the idea that the majority of the DP cells are transcriptionally inactive.

Comparison of level of free and chromatin-bound Pol II separately in control and cKO cells by Western blot discovered increased level of both free and chromatin-bound Pol II (Ser-2p) and Pol II (Ser-5p), while level of both free and chromatin-bound total Pol II was unaffected (Fig. 6k, l). These results are in agreement with results of the Western blot using whole cell lysate (Fig. 1b) and results of the Pol II (Ser-2p) and Pol II (Ser-5p) CUT&Tag experiments (Fig. 6d, g, j, Supplementary Fig. 9e, f), but different from what we have found by total Pol II CUT&Tag showing decreased total Pol II chromatin occupancy under Tox4 deficiency (Fig. 6a, j and Supplementary Fig. 9d) and what we have obtained using K562 cells showing increased level of free but decreased level of chromatin-bound total Pol II[29]. Total Pol II CUT&Tag only identified ~100 genes with significantly decreased occupancy of it upon *Tox4* deletion (Fig. 6a–c and Supplementary Fig. 9d). Therefore, the difference can be explained by the relatively small occupancy change of Pol II (Fig. 6b, c) and the low sensitivity of Western blot. Together, these results suggest that similar to what TOX4 does in K562 cells[29], Tox4 may also facilitate reinitiation in murine DP cells by assisting dephosphorylation of serines 2 and 5 of Pol II CTD.

**Tox4 may also restrict elongation by facilitating Spt5 dephosphorylation in DP cells.** We recently found that besides Pol II CTD Ser-2, TOX4 may also restrict elongation by facilitating SPT5 Thr-806 dephosphorylation by PP1 phosphatases in K562 cells[29]. To test if it is also the case in murine DP cells, we first compared level of phosphorylated Spt5 (p-Spt5) Thr-806 in control and cKO cells. We discovered unaffected level of it upon Tox4 loss (Fig. 7a), which may be due to that the PP2A-Integrator complex is also capable of dephosphorylating Spt5 Thr-806[50]. To determine the effects of Tox4 loss on the chromatin occupancy of Spt5 and p-Spt5 Thr-806, we subsequently performed CUT&Tag experiments using DP cells. Correlation analyses of related biological replicates suggest that the data are highly reproducible (Supplementary Fig. 10a, b).

We found that Spt5 occupancy near TSSs slightly decreased (Fig. 7b, h, Supplementary Fig. 10c, e), which is consistent with the decrease of total Pol II occupancy (Fig. 6a and Supplementary Fig. 8d), p-Spt5 Thr-806 occupancy near TSSs slightly increased, and thus occupancy of p-Spt5 Thr-806 relative to that of Spt5 increased (Fig. 7c, h, Supplementary Fig. 10d, e), which is also in agreement with what we have found in K562 cells[29]. Specifically, the numbers of Tox4-bound genes exhibiting significantly decreased and increased Spt5 occupancies near TSSs were 249 and 93, respectively, while the corresponding numbers near TESs were 127 and 84, respectively (Fig. 7d, e); the numbers of Tox4-bound genes exhibiting significantly decreased and increased p-Spt5 Thr-806 occupancies near TSSs were 4 and 61, respectively, while the corresponding numbers near TESs were 9 and 48, respectively (Fig. 7f, g). Together, these results suggest that similar to what we have found in K562 cells[29], Tox4 may also restrict elongation by facilitating the dephosphorylation of Spt5 Thr-806 by PP1 phosphatases in murine DP cells. Moreover, 2 of the downregulated Tox4 direct targets, *Cd8a* and *Cdkn2d* (Fig. 6j), also exhibited increased occupancy of p-Spt5 Thr-806 (Supplementary Fig. 10e), suggesting that similar to what TOX4 does in K562 cells[29], Tox4 may restrict elongation and facilitate reinitiation of the same gene in murine DP cells (further discussed in the "Discussion" part).

**Tox4 loss affects chromatin accessibility of a small subset of genes.** HMG proteins are capable of modulating chromatin accessibility[51], and Tox1 and 2 have been reported to regulate transcription through modulating chromatin accessibility[31,36]. However, our ATAC-seq analyses in K562 cells suggest that TOX4 is likely to regulate transcription independent of chromatin accessibility modulation[29]. To evaluate if Tox4 regulates chromatin accessibility in murine T cells, we performed ATAC-seq in control and cKO DP cells with high reproducibility (Supplementary Fig. 11a). A set of consensus peaks were obtained by merging peaks from control and Tox4 cKO cells for the identification of regions with accessibility changes (Supplementary Fig. 11b). We found that *Tox4* deletion minimally affects global chromatin accessibility and distribution of accessible sites across genomic features (Supplementary Fig. 11c, d). The numbers of sites with decreased and increased accessibility were 327 and 409, respectively (Supplementary Fig. 11e), and the numbers of Tox4 binding sites with decreased and increased accessibility were 5 and 57, respectively (Supplementary Fig. 11f, g), with fold change > 2 and FDR < 0.05. Moreover, comparative analyses of the Tox4 CUT&Tag, the RNA-seq, and the ATAC-seq discovered only 1 Tox4 direct target gene with significantly changed chromatin accessibility, and comparative analyses of Tox4 CUT&Tag, the TTchem-seq, and the ATAC-seq discovered only 3 Tox4 direct target genes with significantly changed chromatin accessibility. Together, these results suggest that Tox4 is likely to regulate transcription independent of chromatin accessibility modulation in murine DP cells, which is consistent with what we have found in human K562 cells[29].

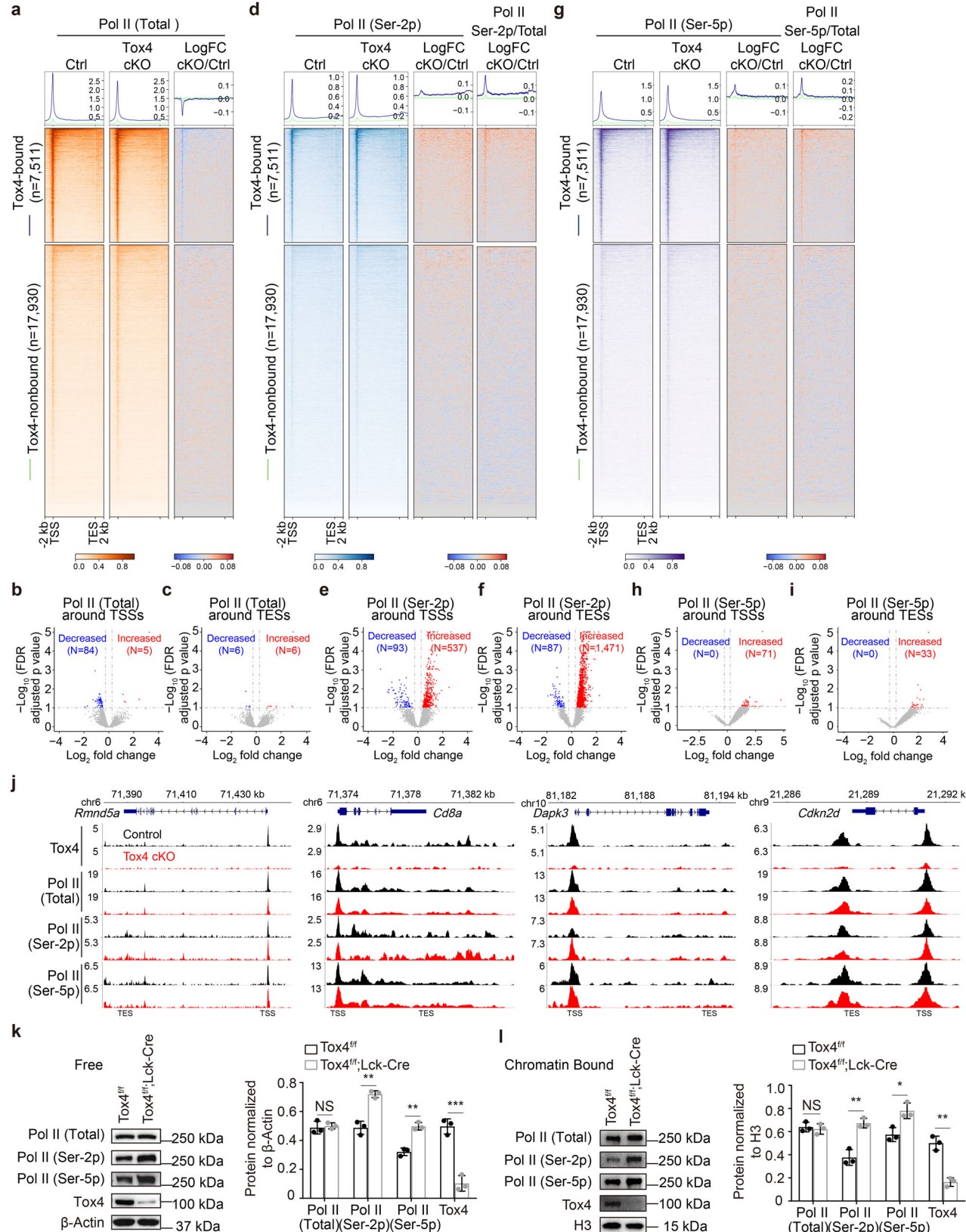

## Discussion

HMG protein TOX4 is one of the regulators of PP1 phosphatases with unknown role in development. In this study, we found that Tox4 conditional knockout in mice partially blocks DN and ISP to DP transition, decreases proliferation of both the DP blast population and CD8, and increases apoptosis of CD8. In addition,

its role in transcriptional regulation is evolutionarily conserved between mouse and human (Fig. 7i).

We recently found that in K562 cells, TOX4, as one of the regulators of PP1 phosphatases, restricts transcriptional elongation by assisting Pol II CTD Ser-2 and Spt5 Thr-806 dephosphorylation and facilitates reinitiation by assisting Pol II CTD

**Fig. 6 Tox4 may restrict elongation but facilitate reinitiation in murine DP cells. a, d, g** Genome-wide meta-gene profiles and heatmaps of CUT&Tag comparing chromatin occupancy of total Pol II (**a**), Pol II (Ser-2p) (**d**), and Pol II (Ser-5p) (**g**) in cKO versus control cells. **b, c** Volcano plots showing Pol II occupancy changes near TSSs (**b**) and TESs (**c**) of Tox4-bound genes in cKO versus control cells. **e, f** Volcano plots showing Pol II (Ser-2p) occupancy changes near TSSs (**e**) and TESs (**f**) of Tox4-bound genes in cKO versus control cells. **h, i** Volcano plots showing Pol II (Ser-5p) occupancy changes near TSSs (**h**) and TESs (**i**) of Tox4-bound genes in cKO versus control cells. **j** Normalized read distribution of CUT&Tag of Tox4, total Pol II, Pol II (Ser-2p), and Pol II (Ser-5p) within the *Rmnd5a*, *Cd8a*, *Dapk3*, and *Cdkn2d* loci in cKO versus control cells. **k, l** Comparison of free (**k**) and chromatin-bound (**l**) Pol II in control and cKO cells by Western blot. Left: representative pictures of Western blot; Right: a bar graph comparing relative level of total Pol II, Pol II (Ser-2p), Pol II (Ser-5p), and Tox4 in control and cKO cells quantified by ImageJ. β-Actin and H3 were the loading controls for free and chromatin-bound proteins, respectively. Pictures in **k** and **l** are representative of three independent experiments (*n* = 3). Statistical significance was determined with a two-sided Student's t-test; the centers and the error bars represent the mean and the SD, respectively. NS: *P* ≥ 0.05, *\*P* < 0.05, *\*\*P* < 0.01, and *\*\*\*P* < 0.001.

serines 5 and 2 dephosphorylation[29]. However, cell lines usually harbor chromatin abnormality, and are known to be several times larger in volume than the corresponding primary cells, suggesting that they are transcriptionally more active than the corresponding primary cells[52] and that there may be major differences between cell lines and the corresponding primary cells with respect to mechanisms of gene regulation. For example, we found that H3K27 trimethylation level in murine erythroleukemic cell line, G1ER, is much lower than that in primary murine erythroid cells from fetal liver[53], and observed that level of H3K27 trimethylation or H2A monoubiquitination in murine megakaryoblastic cell line, L8057, is much lower than that in primary murine megakaryocytes. Therefore, results from cell lines need to be validated using primary cells. In the current study, we found that effects of *Tox4* deletion on the cellular level of total, Ser-2p and Ser-5p Pol II, the chromatin occupancy of total Pol II, and the relative chromatin occupancies of Pol II (Ser-2p), Pol II (Ser-5p) and p-Spt5 Thr-806 greatly resemble those of TOX4 knockout in K562 cells. These results suggest not only that Tox4 restricts elongation and facilitates reinitiation in murine DP cells but also that the roles of TOX4/Tox4 in transcriptional regulation are likely to be evolutionarily conserved. Moreover, our results strongly suggest that Ser-2 of Pol II CTD is also a bone fide target of the PTW/PP1 complex in addition to Ser-5. Actually, one of the previous studies found that WDR82 knockdown increased Ser-2p although the increase of Ser-5p was greater in the cells that they used[28]. Nevertheless, the direct effects of TOX4/Tox4 deficiency on transcription remain to be determined. The AID-TOX4 cell line that we had generated has leaky degradation issue[29], so alternative strategies need to be explored in the future.

With respect to why Tox4 loss only affects transcription of a small subset of genes in DP cells, the reason may be twofold. One is the heterogeneity of DP cells, i.e., DP contains at least three major populations, DPbla, DPre, and DPsel, and the other is the potentially opposing effects of Tox4 on transcription. TOX4/Tox4 is able to facilitate Pol II CTD serines 2 and 5 dephosphorylation. Dephosphorylation of Ser-2 during elongation may restrict binding of elongation and processing factors, whereas dephosphorylation of serines 2 and 5 after termination would facilitate reinitiation. Therefore, TOX4/Tox4 is capable of negatively and positively regulate transcription of the same gene. In addition, TOX4 also facilitates elongation in K562 cells through an unclear mechanism[29]. Consequently, TOX4/Tox4 loss may mainly impair reinitiation and to a lesser degree, elongation[29]. Transcription level of genes is mainly determined by the frequency of reinitiation, so that genes with high expression level are usually downregulated and are more sensitive to TOX4/Tox4 deficiency than those with medium or low expression level (Figs. 4j–l and 5k–n). In contrast, genes with low expression level are the least dependent on reinitiation and the most sensitive to elongation disruption among the three groups, so that some of them are upregulated upon TOX4/Tox4 loss (Fig. 5k–n). However, most of the genes with medium expression level were minimally affected

by TOX4/Tox4 loss because of the opposing effects of TOX4/Tox4 on their transcription.

Tox1 and 2 are critical regulators of the immune systems. In T cell development, TOX activates *Zbtb7b* (encoding ThPOK) transcription and induces the full CD4 lineage gene program with unclear mechanisms[16,17]. Tox2 is dispensable for T cell development but critical for driving transcription of $T_{fh}$-associated genes in an HMG box (chromatin accessibility modulation) dependent manner[31]. In contrast, we found in the current study that Tox4 knockout impairs proliferation of the highly proliferating T cell subpopulations, DP blast 1 and 2, by decreasing transcription of highly expressed genes, including key regulators of G2 and M phases of the cell cycle and genes of TCR signaling. These results raise a possibility that highly proliferating cells are more dependent on TOX4/Tox4 than low proliferating ones in development and diseases. Future works are needed to further investigate this matter.

Tox family transcriptional regulators contain HMG box, which is capable of modulating chromatin accessibility[32]. Among the family members, Tox1 and 2 has been found regulating transcription and development through modulating chromatin accessibility. For example, *Tox1* deletion in murine CD8 cells affects chromatin accessibility of around four thousand sites[30], and Tox2 overexpression affects chromatin accessibility of over eight thousand sites in $T_H0$ and $T_{FH}$ like cells, respectively[31]. We found in the current study that *Tox4* deletion in murine DP cells only significantly affected chromatin accessibility of 736 sites, and among them, only 8.5% (63 out of 736) may be directly regulated by it. In addition, only 1.2% (1 out of 82) of the direct targets of Tox4 identified by CUT&Tag and RNA-seq and 1% (3 out of 280) of the direct targets identified by CUT&Tag and TTchem-seq exhibited significant chromatin accessibility change upon Tox4 loss. These results suggest that, unlike what Tox1 and 2 do in murine T cells but similar to what TOX4 does in K562 cells, Tox4 mainly regulates transcription though PP1 phosphatases other than modulating chromatin accessibility.

## Methods

**Mice.** *Tox4* floxed mice were generated by CRISPR-Cas9 mediated knock-in of floxed exons 4–6 into fertilized eggs from mice of C57BL/6 background in collaboration with Shanghai Model Organisms, Inc. Lck-Cre mice were purchased from Shanghai Model Organisms, Inc[54], and Vav-iCre mice were purchased from Cyagen[55]. All studies were performed on gender-matched littermate mice 6–8 weeks of age, Mice were bred and maintained in pathogen-free facilities at Shanghai Jiao Tong University and studies were conducted in accordance with the Regulations of Shanghai Jiao Tong University Animal Studies Committee.

**Flow cytometry.** Bone marrow cells, thymocytes, and lymphocytes were washed and resuspended in PBS plus 1% fetal bovine serum (FBS) at a concentration of $1–5 \times 10^6$/ml. All the staining were performed on ice unless stated otherwise. For surface protein staining, cells were stained with anti-CD16/32 antibodies for 10 min to block non-specific binding to Fc receptors before staining with antibodies targeting surface proteins of interest. Intracellular staining for Ki67 was performed using the Foxp3/Transcription Factor Staining Buffer Set (eBioscience, cat. no. 00-5523-00). For Annexin V staining, cells were rested in

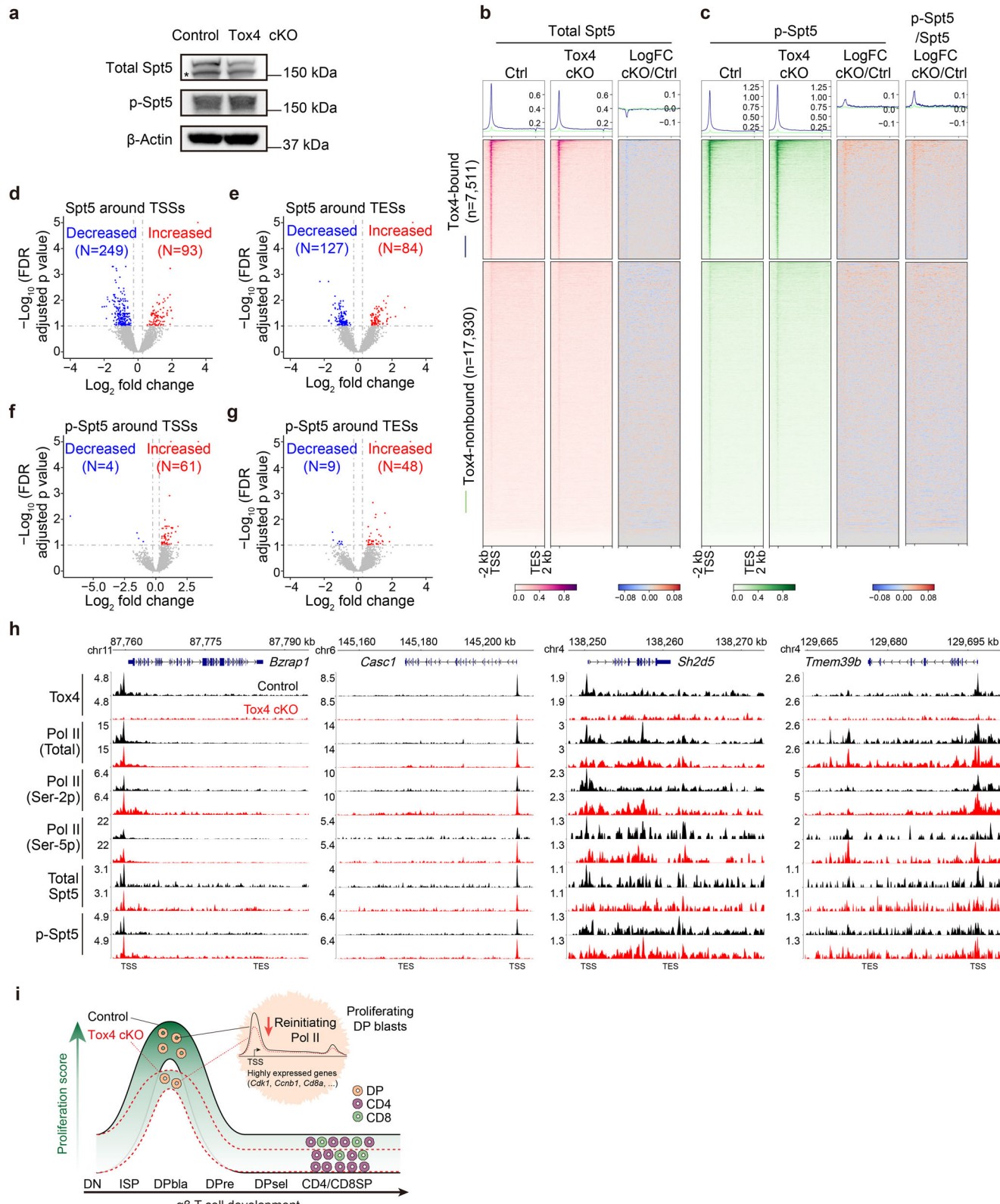

**CUT&Tag and data analyses**. CUT&Tag experiments were performed as previously described[29,56]. Briefly, 250,000 murine cells were used for each experiment. Cells were bound to Concanavalin A-coated beads without fixation and chromatin opening. After primary and secondary antibodies binding, pA-Tn5 transposome binding, and tagmentation, DNA was extracted and amplified by PCR. Antibodies used for CUT&Tag are listed in Supplementary Table 2. Raw reads were filtered using fastp (version 0.13.1, default parameters)[57] and aligned to mouse genome mm10 using Bowtie2 (version 2.3.4.1)[58]. Low-quality

RPMI-1640 supplemented with 10% FBS for 5 h at 37 °C before stained with an anti-Annexin V antibody in Binding Buffer (0.01 M HEPES, 0.14 M NaCl, and 2.5 mM CaCl$_2$). 7-Aminoactinomycin D (7-AAD) was added to stain dead cells. All the flow cytometry analyses were performed on a CytoFLEX flow cytometer (Beckman Coulter). Sorting of DN, ISP, DP, DP bast, CD4, and CD8 cells for Western blot, RNA-seq, or qRT-PCR was performed on a BD FACSAria III sorter. The antibodies used for flow cytometric analyses are listed in Supplementary Table 1.

**Fig. 7 Tox4 may also restrict elongation by facilitating Spt5 dephosphorylation in murine DP cells. a** Comparison of cellular level of Spt5 and p-Spt5 Thr-806 by Western blot in control and cKO cells. The non-specific band is highlighted by a "*". β-Actin was used as a loading control. **b, c** Genome-wide meta-gene profiles and heatmaps of CUT&Tag comparing chromatin occupancy of Spt5 (**b**) and p-Spt5 Thr-806 (**c**) in cKO versus control cells. **d, e** Volcano plots showing Spt5 occupancy changes near TSSs (**d**) and TESs (**e**) of Tox4-bound genes in cKO versus control cells. **f, g** Volcano plots showing p-Spt5 Thr-806 occupancy changes near TSSs (**f**) and TESs (**g**) of Tox4-bound genes in cKO versus control cells. **h** Normalized read distribution of CUT&Tag of Tox4, total Pol II, Pol II (Ser-2p), Pol II (Ser-5p), Spt5, and p-Spt5 Thr-806 within the *Bzrap1*, *Casc1*, *Sh2d5*, and *Tmem39b* loci in cKO versus control cells. **i** A schematic illustration of the role of Tox4 in T cell development. During T cell development, Tox4 regulates the transition from DN and ISP to DP, the proliferation of the highly proliferating DP blast population, and the homeostasis of CD8 through restricting elongation and facilitating reinitiation of target genes.

alignments were filtered out using SAMtools (version 0.1.19)[59] with command "samtools view -F 1804 -q 25". MarkDuplicates tools in Picard (https://broadinstitute.github.io/picard/) was used to identify and remove PCR duplicates from the aligned reads. Peak calling was performed using SEACR (version 1.3)[60] with an FDR threshold 0.02 in stringent mode. Peaks were called initially from merged reads of two biological replicates, and among them, those cannot be called subsequently from either of the biological replicates were removed. The remaining peaks were defined as high confidence ones. A gene is considered as bound by one factor if any peak of this factor is found from 2 kb upstream to 300 bp downstream of this gene. EdgeR[61] was used to analyze differential occupancy around TSSs (from 2 kb upstream to 2 kb downstream of TSSs) or TESs (from 2 kb upstream to 5 kb downstream of TESs), and genes with a BH-adjusted *p* value < 0.05 and fold change ≥ 1.2 were identified as differentially bound.

**RNA extraction, reverse transcription, RNA-seq, and data analyses**. RNA was extracted from cells using TRNzol Universal Reagent (TIANGEN, cat. no. DP424) or Quick-RNA MiniPrep Kit (Zymo Research, cat. no. R1054) by following the manufacturers' protocols. The primers used for quantitative reverse transcription polymerase chain reaction (qRT-PCR) are listed in Supplementary Table 3. Libraries of strand-specific RNA-seq were constructed as previously described[62]. Raw reads were filtered using fastp[57] (version 0.13.1, default parameters) and mapped to mm10 for mouse DP thymocytes, using HISAT2[63] (version 2.1.0) with parameters "--rna-strandness RF –dta". Read counts per gene were calculated in strand-specific manner using featureCounts[64]. Differential expression analysis was performed using DESeq2[65], and genes with mean TPM ≥ 1, FDR < 0.05, and fold change > 1.5 were identified as significantly differentially expressed.

**Cell purification**. Thymi and lymph nodes were harvested from 6- to 8-week-old mice and gently passed through a 70-μm cell strainer (BD, cat. no. 352350). DP cells were purified from thymus using Dynabeads FlowComp Mouse CD4 Kit (Invitrogen, cat. no. 11461D). CD4 T cells were purified from lymph nodes using CD4$^+$ T cell isolation Kit (Miltenyi Biotec, cat. no. 130-104-454). CD8 T cells were purified from lymph nodes using CD8a$^+$ T cell isolation Kit (Miltenyi Biotec, cat. no. 130-104-075). Purified cells were used for CUT&Tag or ATAC-seq experiments.

**T cell activation and CFSE assay**. Sorted T cells were cultured with T cell medium (RPMI-1640, 10% FBS, 10 mM HEPES, 1% penicillin/streptomycin, 1 mM sodium pyruvate, 50 mM β-mercaptoethanol and 2 mM l-glutamine), and activated by plate-bound α-CD3 (Biolegend, 100339, Clone 145-2C11, coated overnight, 3 mg/ml) and α-CD28 (Biolegend, 102115, Clone: 37.51, coated overnight, 3 mg/ml). For the analysis of activation, T cells were collected 12 and 36 h after activation for FACS. CFSE assays were performed using the CellTrace™ CFSE Cell Proliferation Kit (Invitrogen, C34570). Cells were initially incubated in PBS containing 5 mM CFSE at room temperature for 5 min, then cultured in T cell medium without CFSE, and finally analyzed by FACS 24, 48, and 72 h later. Only results of 72 h are shown.

**Single-cell RNA-seq and data analyses**

*Single-cell RNA-seq.* Thymocytes in suspension were counted after passing through a 70-μm cell strainer (BD, cat. no. 352350). Single cells were captured in droplet emulsions using a Chromium Controller (10X Genomics), and 20,000 cells were used for each experiment. scRNA-seq libraries were constructed by following the 10X Genomics protocol using Chromium Single Cell 3′ Reagent Kits v3.

*Single-cell RNA-seq alignment and quantification.* scRNA-seq reads were aligned to the GRCm38 (mm10) reference genome and quantified using 'cellranger count' (10x Genomics, version 6.1.1) with default parameters. scRNA-seq UMI count matrices were imported to R 4.1.0 and gene expression data analysis was performed using the R/Seurat package (version 4.0.4)[66].

*Data quality control.* Genes expressed in fewer than three cells and cells with less than 200 genes were removed when import data to R using Seurat package. Cells were further filtered by removing both UMI count outliers (top and bottom 2.5%)

and those with mitochondrial read count exceeding 10% of the total. Moreover, doublets were identified separately for every sample using DoubletFinder (version 2.0.3)[67], and 388 to 844 cells were removed from the four samples, respectively.

*Normalization and data integration.* Before further analyses, sample was normalized individually using the SCTransform function of Seurat, setting parameter 'vars.to.regress' as the percentage of mitochondrial genes and other parameters as default. After normalization, data of 4 samples were integrated on the basis of 3000 most variable features identified by the SCTransform function of Seurat. For the integrated data, principal component analysis (PCA) was performed for initial reduction in the dimensionality, and UMAP analysis was performed on the basis of the first 30 dimensions. The cells were clustered using the FindClusters function initially with a resolution of 0.5, and 21 clusters were obtained. One cluster with high mitochondrial gene expression was removed afterward, and cell clustering was reperformed with a resolution of 0.4, which resulted in 20 clusters.

*Cell cluster annotation.* The 20 thymocyte clusters were annotated according to both the canonical markers[41] and the automated annotation result generated by the singleR (version 1.6.1) package[68] with the Immunological Genome Project (ImmGen) reference data.

*Identification of cluster markers.* To identify markers of each cluster, differential expression (DE) tests were performed using FindMarkers/FindAllMarkers functions in Seurat with Wilcoxon rank-sum test. Genes with log-fold differences > 0.25, expression in at least 10% of the cells of a test group and Bonferroni-corrected *P* values < 0.01 were considered as significantly differentially expressed genes (DEGs). Cluster markers are highly expressed genes of each cluster identified by DE tests between one cluster and the rest of clusters, and top-ranked genes (by log-fold differences) from each cluster were extracted for further illustration.

*Differential expression tests.* To identify genes whose expression were affected by *Tox4* depletion in each cluster, conserved differentially expressed genes between control and cKO cells in each cluster were identified with the FindConservedMarkers function in Seurat. Only those with log-fold changes > 0.25, combined *P* values < 0.05, expression in at least 10% of the cells of a test group and changes in the same direction in both replicates are regarded as significantly DEGs.

*Pseudotime analysis.* Trajectory analysis was performed with Monocle v3[69] and pseudotime for each cell was estimated. Specifically, a CellDataSet object was constructed for cells in all the 15 αβ T cell clusters, and preprocessed using the preprocess_cds function in Monocle. UMAP dimensionality reduction and cell clustering were performed afterwards followed by replacing the UMAP dimensions calculated by Monocle with those calculated by Seurat as above mentioned for the purpose of consistency. Finally, trajectory analysis was performed with the learn_graph function and pseudotime was predicted with the order_cells function with the root_pr_nodes set as cells in DN cluster. The trajectory and the pseudotime were visualized using plot_cells.

*Proliferation score calculation.* Proliferation scores were calculated by averaging expression of known proliferation-related genes[70] using the AverageExpression function in Seurat. The proliferation-related genes are *Aurka*, *Bub1*, *Ccnb1*, *Ccnd1*, *Ccne1*, *Dek*, *Fen1*, *Foxm1*, *H2afz*, *Hmgb2*, *Mcm2*, *Mcm3*, *Mcm4*, *Mcm5*, *Mcm6*, *Mki67*, *Mybl2*, *Pcna*, *Plk1*, *Top2a*, *Tyms*, and *Zwint*. The difference of proliferation score between control and cKO cells in each cluster was tested using two-sided Wilcoxon rank-sum test.

*Cell cycle phase analysis.* Cells were assigned to either 'G2/M' phase or 'S' phase using the CellCycleScoring function of the Seurat package according to the previously defined cell cycle genes specific to either G2/M phase or S phase[71]. Cells expressing none of the genes were assigned to the 'G1' phase.

**ATAC-seq and data analyses**. Expression and purification of Tn5 transposase and transposome assembly were conducted as previously described[72]. ATAC-seq experiments were performed by following a published protocol[73]. Briefly, 100,000

cells were used for each experiment. After nuclei preparation, tagmentation, termination, and DNA purification, each sample was amplified by PCR with one universal forward primer and one of the reverse primers with different indexes.

ATAC-seq pair-end reads were filtered using fastp (version 0.13.1, default parameters)[57] and aligned to the mouse genome mm10 using Bowtie2 (version 2.3.4.1)[58] with parameter "-X 2000". SAMtools was used to filter reads to keep only those mapped to Chr1-22 and ChrX, and MarkDuplicates tool from Picard was used to identify and remove PCR duplicates from the aligned reads. The final deduplicated BAM file was used in the downstream analyses.

Tn5 transposase insertions, which refer to the precise single-base locations where Tn5 transposase accessed the chromatin, were identified by correcting the read start positions by a constant offset ("+" stranded +4 bp, "−" stranded −5 bp). To generate depth-normalized accessibility tracks, bigwig files were constructed based on the Tn5 offset-corrected insertion sites using GenomicRanges[74] and rtracklayer[75] packages in R. Meta-gene profile plots were generated using computeMatrix and plotProfile from deepTools[76]. For each replicate, peak calling was performed on the Tn5-corrected single-base insertions using the "MACS2 callpeak" command with parameters "-g hs -q 0.01 --shift -19 --extsize 38 --nomodel --nolambda --keep-dup all --call-summits". The peaks were then filtered to remove peaks overlapping the mm10 blacklisted region. Peaks initially were called from merged reads of two biological replicates, and among them, those cannot be called from either of the biological replicates were removed afterward. The remaining peaks were defined as high confidence ones. A consensus peak set was obtained by merging the high confidence peaks identified in each cell type using mergeBed. Peak annotation of high confidence peaks of each cell type and the final consensus peak set was performed using ChIPseeker (version 1.18.0)[77] package in Bioconductor.

Tn5 transposase insertion count matrix was constructed by counting Tn5 transposase insertions in each consensus peak in every sample, and was taken as input for edgeR[61] to perform differential accessibility analysis. Consensus peaks with FDR-adjusted P value < 0.05 and fold change ≥ 2 were defined as differentially accessible peaks upon Tox4 loss, and differentially accessible Tox4-binding sites are those differentially accessible peaks with Tox4 occupancy. Accessibility changes of direct target genes of Tox4 were determined by comparative analyses of CUT&Tag, RNA-seq, and ATAC-seq data of Tox4.

**TTchem-seq and data analyses.** TTchem-seq experiments were performed as previously described[43,44] with minor modifications. Briefly, $1.5 \times 10^7$ cells were used for each experiment; cells were transferred to fresh antibiotics-free medium and cultured for 0.5 h before 4-thiouridine (4sU) treatment; total RNA was extracted using TRNzol according to the manufacturer's instructions; 1 μg 4-thiouracil (4TU) labeled *S. cerevisiae* BY4741 RNA was added to 100 μg total mouse RNA as spike-in; after fragmentation, biotinylation of 4sU-labeled RNA, purification of biotinylated RNA with Dynabeads M-280 streptavidin and rRNA depletion, strand-specific RNA-seq libraries were constructed as previously described[62].

Analyses of TTchem-seq data were performed as previously described[43] with minor modifications. Specifically, raw reads were filtered using fastp (version 0.13.1, default parameters)[57]. For target (*Mus musculus* GRCm38) or spike-in (*S. cerevisiae* sacCer3) genome, genome sequences and RefGene annotation file were downloaded from UCSC, and STAR genome index was prepared using STAR (version 2.7.9a)[78] with the "--runMode genomeGenerate" option. Filtered reads were aligned against each index using STAR with the "-quantMode GeneCounts" option. SAMtools (version 0.1.19)[59] and Picard were used to sort, index, and mark duplicate reads in the resulting genome BAM files.

We and others have found that spike-in in different samples frequently got sequenced without equal chance for unknown reason. Although spike-in was added to every TTchem-seq sample, normalization by spike-in was not applied to our dataset because they did not show global changes before using spike-in to manifest the global change. Mouse gene count matrix was constructed from the STAR output files. Differential expression analysis was performed with Bioconductor DESeq2 package[65], and genes with mean CPM ≥ 1, adjusted P value < 0.05 and fold change ≥ 1.2 were identified as significantly differentially expressed. To create sense or antisense meta-profiles for gene-body regions of protein-coding genes, mate 2 reads in the BAM files were selected using SAMtools and passed to Ngs.plot using the "-SS" option.

**Separation of free and chromatin-bound proteins.** Cells were collected by centrifuging at 4000 rpm for 5 min at 4 °C, resuspended in nucleus lysis buffer (20 mM Tris-HCl, pH 7.5, 3 mM EDTA, 10% glycerol, 150 mM potassium acetate, 1.5 mM MgCl₂, 1 mM DTT, 0.1% NP-40, 1 mM PMSF, and protease inhibitors) to $1 \times 10^8$ cells/ml, homogenized in a 2-ml Teflon Dounce homogenizer for 60 times, and centrifuged at $15,000 \times g$ for 10 min at 4 °C. The supernatant (the free fraction) was transferred to a new tube. The pellet was resuspended in 1 × SDS-PAGE loading buffer to the equivalent of $1 \times 10^8$ cells/ml, sonicated for 1 h at 4 °C using Bioruptor Pico (Diagenode), centrifuged at $15,000 \times g$ for 10 min at 4 °C, and the supernatant (the chromatin-bound fraction) was transferred to a new tube. The free and the chromatin-bound fractions were analyzed by Western blot.

**Statistics and reproducibility.** No statistical methods were used to predetermine sample size. The experiments were not randomized, and investigators were not blinded to allocation during experiments and outcome assessment.

All statistical tests were two-sided unless otherwise stated. All bar graphs are representative of three or more independent experiments as indicated in the figure legends. Statistical significance was determined with a two-sided Student's t-test; the centers and the error bars represent the mean and the SD, respectively. Where P values are reported, an alpha level < 0.05 was considered statistically significant. Two-sided Student's t-test was performed using GraphPad Prism (version 7). Wilcoxon rank-sum test was performed by R.

The Benjamini–Hochberg (BH) correction method was used to adjust the P values where multi-testing corrections were involved. FDR-adjusted P values and fold changes (FCs) for expression changes were derived from DESeq2 analysis[65]. P values and FCs for accessibility changes were derived from edgeR[61] analysis, and P values were adjusted as mentioned above. Reproducibility of two biological replicates of CUT&Tag or ATAC-seq data were assessed using Pearson correlation coefficient calculated by deepTools. FDR-adjusted P values for GO terms enrichment were derived from GO analysis by clusterProfiler.

**Reporting summary.** Further information on research design is available in the Nature Portfolio Reporting Summary linked to this article.

## Data availability
Next-generation sequencing data have been submitted to GEO repository under accession number GSE190041. Gating strategy for FACS that were not included in the main Figures are available in Supplementary Figs. 12–14. The uncropped images of Western blot experiments are available in Supplementary Figs. 15–20. The source data of Figures are provided in the file Supplementary Data 1. All other data are available from the corresponding author upon reasonable request.

## Code availability
Details of publicly available software used in the study are given in the "Methods". No custom code or mathematical algorithm that is deemed central to the conclusions was used.

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

## Acknowledgements

The authors would like to thank N. Xiao (XMU) for critical reading of the manuscript, R. Fisher (MSSM) for the p-SPT5 Thr806 antibody, H. Jiang (SJTU) for technical support for cell sorting, and W. Chen (SJTU) for advice on analyzing T cell activation and proliferation. M.Y. is supported by grants from The Program for Professor of Special Appointment (Eastern Scholar) at Shanghai Institutions of Higher Learning, and National Natural Science Foundation of China (32270589).

## Author contributions

M.Y., T.W., and Z.L. designed the experiments, T.W., J.Z., Z.L., and Z.Y. performed the experiments and analyzed the data, R.Z., A.W., and W.W. performed the bioinformatics analysis, and M.Y., T.W., A.W., L.J., and T.N. wrote the paper.

## Competing interests

The authors declare no competing interests.
