## [Peer Review File · Communications Biology]

Reviewers' comments:

Reviewer #1 (Remarks to the Author):

The manuscript "Tox4 is an evolutionarily conserved regulator of elongation and recycling of RNA polymerase II" by Liu and colleagues identifies Tox4 – a member of the PTW RNA Polymerase II (RNAPII) phosphatase – as relevant mediator of T cell development. The experiments are overall well conducted and scientifically sound, but some aspects raise concerns that need to be addressed.

1. The reduced cellularity of Tox4 KO thymi is very striking. The mechanisms behind this are quite confusing, being based on a mixture of increased cell proliferation and apoptosis of the involved populations. In detail, the authors suggest that the reduced cellularity can be due to increased apoptosis (line 168), but this looks negligible – albeit mildly significant – when compared to increased Ki67. Even the strongest apoptosis increase (CD8, 2% to 7.8%) appears unlikely to explain reduced cellularity when compared to Ki67-positive cells increase (37% to 43%).

2. Tox4 CUT&Tag analysis looks a bit problematic.

- I find the metagene comparison of Tox4-bound and -nonbound genes a bit unusual. Why not simply plotting the usual metagene representative of all genes?
- Additionally, what is the negative control there? There should be a no/IgG-antibody control?
- Why is Tox4 cKO displaying still a TSS enrichment? Tox4 deletion is expected to be complete, isn't it?
- Previous studies (Austena et al 2015, Cortazar et al 2019, Cossa et al 2020) identified/suggested the presence of the PTW phosphatase also at the TES, where it is relevant for termination. Why is TOX4 not detected there by CUT&Tag?

3. RNAPII CUT&Tag experiments technically raise similar questions.

- A metagene plot for all genes including Ctrl, Tox4 cKO and a negative control plot should be shown.
- Additionally, the Tox4-bound/nonbound stratification appears particularly misleading for this specific RNAPII binding analysis. For example, it shows that Tox4-nonbound genes have no RNAPII, suggesting that Tox4 promotes RNAPII binding, which doesn't appear to be the case when looking at the Tox4 cKO metagene. The authors should solve this apparent discrepancy.
- Along the same line, is Tox4 binding stratifying to RNAPII-bound genes? Or to expressed genes? Proper correlation analyses should be performed.
- Data supporting the claim that Tox4 loss induces loss of RNAPII occupancy are very weak, both in the metagene and in the shown genome browser tracks. I think the actual relevant phenotype is the increase of pSer2-RNAPII along the gene. Still, this is probably indirect, as the proposed target of the PTW phosphatase is pSer5-RNAPII. For this reason, I think the authors should perform also pSer5-RNAPII CUT&Tag.

4. The authors mention several times "transcription reinitiation" (line 314 and elsewhere), but this term doesn't recall any standard description of RNAPII transcription behavior. Do they mean transcription elongation after pause-release (= "late elongation")? Or RNAPII eviction from pause site (= "premature termination") and RNAPII reloading at TSS? The concept should be extensively explained in the introduction or where relevant and, more importantly, well supported by data.

5. Concerning the extragenic transcription analysis.

- Why are the outputs of RNA-Seq and TT-Seq so different? This could happen, but I find it extremely surprising that the most striking effects are identified by RNA-Seq and not TT-Seq (which should better identify unstable transcripts, as these extragenic transcripts usually are).
- Why are the TT-Seq results commented in the "take-home" sentence of the paragraph and not the

more striking RNA-Seq ones?

- Why are only the up-regulated transcripts analyzed?
- Some gene browser track examples here would help visualizing the phenotype.

6. The authors repeatedly refer to a revised manuscript to back or comment their findings. These referrals are impossible to evaluate, so the authors should either remove them or describe better the "K562 phenotype" they refer to. On the other side, most of the mechanistic analyses appear to be a confirmation of that revised/unpublished manuscript, which somehow diminishes the actual novelty of the findings reported here.

7. Generally speaking, why is a key regulator of RNAPII transcription (=binding over 7,500 genes) mediating the up-/down-regulation of only very few of them? I believe this could be in part due to inherent secondary compensation effects upon TOX4 deletion. To investigate this hypothesis, at least some of the TOX4-dependent transcription events should be assessed upon shorter-term depletion methods.

Minor points.

- Multiple studies about the PP1-PNUTS-TOX4-WDR82 complex (including the original Lee et al 2010) refer to it as the "PTW phosphatase" (or PP1/PTW) complex. I would recommend the authors to use this term, instead of the "PP1C" abbreviation, which usually refers to PP1 catalytic subunits.
- The authors should clarify that all scatter plots in Figures 1 and 2 are representative experiments, otherwise the accompanying bar charts would look discrepant.
- The authors mention Figures "S3a and S3b" which don't exist (line 176). Similarly, also Figure "S7a-d" and "S8a-d" are miscalled (lines 299, 332).
- What is the exact take-home message of Figure 3e? The authors comment it as a proof of "no effect of global transcription output", but what it mostly shows is that there's relevant variability between replicates. How were the experiments/replicates normalized?
- For clarity, the legend of Figure 3h could rather label RNASeq and TTSeq as "(mature) mRNA level" and "nascent mRNA level", respectively.
- What are Tox4-bound genes (and/or the ones up-/down-regulated upon Tox cKO) doing? A GO Term analysis should be attempted.
- I think the single-cell analysis part could be moved before the Tox4 CUT&Tag part. This would help the overall flow of the manuscript.
- The decrease of Cdk1, Ccnb1, Ccnb2 in the DPbla2 population doesn't appear very striking. Is there some supporting statistics here? This should also be validated by alternative means (e.g., qPCR).
- The passage about the overexpression of ribosomal protein genes (line 278) is speculation with scarce relevance for the rest of the story, so it should be probably omitted.

Reviewer #2 (Remarks to the Author):

In this study, Liu and colleagues report the role of Tox4, a regulatory subunit of protein phosphatase 1 (PP1), in T cell development in mice by controlling the expression of the genes involved in the T cell receptor (TCR) signaling pathway. From their mouse genetic and functional genomics data, the authors also suggest that Tox4 regulates transcriptional elongation and reinitiation.

The experiments have been designed elegantly and thoughtfully. The language is lucid, and overall, the paper is written quite well. However, some major and minor concerns need to be addressed before publication.

Major Comments:

1. Authors suggest that the loss of Tox4 impairs T cell development in part by downregulating TCR signaling genes, e.g., Cdk8a. Then based on their genomics data—the genome-wide distribution of Pol II, Spt5, Spt5-pThr806—the authors conclude that Tox4 functions as a suppressor of elongation. How can the authors reconcile these two seemingly contradictory conclusions?

2. Fig 1b – There is a difference between the increase in Pol II CTD-pSer2 and -pSer5 upon depletion of Tox4. The increase is more pronounced in Pol II-pSer2—maybe due to the low level of pSer2 in control cells—despite the equal protein loading (as observed by Actin). An explanation for the differences would be beneficial for the reader.

3. Undoubtedly, it is well accepted that the Pol II CTD Ser2 phosphorylation accumulates beyond the polyadenylation site (PAS, authors defined as TES). The pSer2 distribution by CUT&Tag in Fig 5d and 5g resembles Pol II and/or Pol II CTD pSer5 distribution—peaks around the TSS with almost no accumulation of the same beyond the PAS. Therefore, it is recommended to check the anti-pSer2 antibody. Moreover, the observed non-canonical distribution of pSer2 puts all of the data in Fig 5 related to pSer2 in question. This reviewer suggests redoing the experiments to identify the canonical distribution of pSer2 in control and Tox4 KO cells and then analyzing accordingly.

4. How does the author conclude that Tox4 may also facilitate transcriptional reinitiation by measuring Pol II and Pol II pSer2 (though it is not right) occupancy on chromatin? The level of non-chromatin-bound hypophosphorylated Pol II critically controls initiation. Therefore, the authors should measure the level of non-chromatin bound (free) pSer2 and/or pSer5 in both cytoplasmic and nuclear fractions in control and Tox4 KO conditions.

5. While the authors explain the decrease and increase in Pol II, Spt5, and Spt5-pThr806 occupancy near the TESs, does it upstream, around, or beyond the TESs (PASs)? Moreover, it would be beneficial for the readers if the authors could mark the TES in the schematic diagrams of the example genes in the browser tracks.

Minor Comments:

1. Some minor typographical errors and spelling mistakes need to be rectified. For example, on page number 8, "To facilitates" would be "To facilitate".

2. Authors can consider rephrasing and/or shortening some sentences. That way, readers would be more invested while reading. On the other hand, bigger sentences might distract the readers.

3. Fig 1d. and 1f. – Both the "% of DP cells" and "cell number of DP cells" are increasing compared to the DN cells. The authors probably refer to the decrease in the conditional KO DP cells in both the figures compared to the control cells. It would be beneficial if the authors rephrase or reconstruct the sentence describing Fig 1c, d, and f.

Reviewer #3 (Remarks to the Author):

While the role of TOX and TOX2 HMGbox proteins in development and functioning of the immune system has been the focus of much work, nothing has been reported in regard to the in vivo role of TOX4 in lymphocyte development. Using a conditional knockout model, this manuscript describes a very comprehensive and well performed analysis of a role for TOX4 in the thymus, in part based on somewhat similar results that have been obtained in previous work on human lymphoblast CML K562 cells.

Overall, the work is of high quality, has appropriate statistical analyses, uses state-of-the-art available technologies, and the findings novel. The biggest limitation in terms of significance is the relatively modest effect of loss of TOX4. This is somewhat surprising when compared to loss of other TOX family members in certain immune (and other) contexts, where loss of these nuclear proteins can have profound effects on cell differentiation/lineage commitment. The authors repeatedly state that one reason for this is the heterogeneity of double positive thymocytes. It is not at all clear to this reviewer what heterogeneity is being referred to and how this explains the results. This should be clarified. And it is all the more surprising given the profound enhancement of Pol II ser-2P on bulk cells.

In this regard is it also possible that TOX4 plays a modest accessory function in the thymus but has a more profound effect on mature T cells? Do the KO T cells in these mice proliferate normally in response to TCR stimulation (Fig. 2 only shows steady state LN T cells)? It is also stated that TOX4 unlike other family members likely regulates transcription in an HMG-box-independent manner because of limited observed chromatin accessibility changes. The logic here is not clear nor supported by data. As the HMG-box is the DNA-binding domain one would expect that it likely plays some role in targeting the protein (along with presumably distinct protein binding partners) to the various CUT&TAG sites identified. And given the strong conservation of this sequence it seems likely it also is key to TOX4 function, regardless of the ultimate mechanism of action in regulating transcription.

Other issues:

Figure 1b has a band in the TOX4 blot. Is this TOX4 or a nonspecific band? If the latter this needs to be documented. Of course, reduced efficiency of TOX4 deletion could also explain the relatively modest effect. Can the KO make an in-frame truncated protein (would it be detected by the antibody)?

Controls throughout are fl/fl mice. Just the most basic finding in terms of % and numbers of DPs should be presented for control cre+ mice to eliminate potential artifacts caused by overexpression of cre (which has been noted for some strains).

Minor: It is stated that the cre mice are lck distal promoter Is that a typo as lck proximal promoter expression is usually earlier?

Reviewers' comments:

Reviewer #1 (Remarks to the Author):

The manuscript "Tox4 is an evolutionarily conserved regulator of elongation and recycling of RNA polymerase II" by Liu and colleagues identifies Tox4 – a member of the PTW RNA Polymerase II (RNAPII) phosphatase – as relevant mediator of T cell development. The experiments are overall well conducted and scientifically sound, but some aspects raise concerns that need to be addressed.

We'd like to thank the Reviewer for the valuable advice and suggestions. We have performed all the required experiments and revised the manuscript by following the Reviewer's advice. Additionally, we have changed the title of the manuscript to emphasize its relevance to Developmental Biology.

1. The reduced cellularity of Tox4 KO thymi is very striking. The mechanisms behind this are quite confusing, being based on a mixture of increased cell proliferation and apoptosis of the involved populations. In detail, the authors suggest that the reduced cellularity can be due to increased apoptosis (line 168), but this looks negligible – albeit mildly significant – when compared to increased Ki67. Even the strongest apoptosis increase (CD8, 2% to 7.8%) appears unlikely to explain reduced cellularity when compared to Ki67-positive cells increase (37% to 43%).

We'd like to thank the Reviewer for pointing this out, and agree with the Reviewer that it is hard to conclude that the reduced cellularity is caused by increased apoptosis. In the initial submission, we analyzed 3 pairs (n=3) of mice to generate Figure 2. In the revised manuscript, we have analyzed 2 more pairs of mice (n=5). The conclusions are almost the same, but increased apoptosis of CD8⁺ cells upon *Tox4* deletion has become more striking. The causes for decreased thymic cellularity upon *Tox4* deletion are likely to be partial developmental block from the double negative (DN) to the double positive (DP) stage (Fig. 1c-g), increased apoptosis of CD8⁺ cells (Fig. 2e-g), and decreased proliferation of the DP blast population within DP cells (Fig. 4d-i).

Fig. 2e-g. **e** Representative plots of flow cytometric analyses of 7-AAD and Annexin V staining for apoptosis in thymocyte. **f** Bar graphs comparing frequency of early apoptotic (Annexin V+7-AAD-) cells of T cell subpopulations in the thymus in control and cKO mice. **g** Bar graphs comparing frequency of late apoptotic (Annexin V+7-AAD+) cells of T cell subpopulations in the thymus in control and cKO mice. Data shown are the mean \pm S.D. of 5 independent experiments. Statistical significance was determined with a two-sided Student's t-test. NS: $P \geq 0.05$ and *** $P < 0.001$.

2. Tox4 CUT&Tag analysis looks a bit problematic.

- I find the metagene comparison of Tox4-bound and -nonbound genes a bit unusual. Why not simply plotting the usual metagene representative of all genes?

We used to generate metagene plots for all genes. However, we were frequently asked by Reviewers recently (since 2020) to generate metagene plots of bound genes only so that we generate metaplots for bound and non-bound genes, respectively. In the revised manuscript, we have also generated metagene plots of all genes for all the CUT&Tag experiments, and placed them in the corresponding supplementary Figures.

- Additionally, what is the negative control there? There should be a no/IgG-antibody control?

Isotype matched IgG control was included for each group of our CUT&Tag experiments but was not sequenced because of the low quantity of DNA immunoprecipitated by IgG (Please see the attached Figure R1 blow.). Additionally, we'd like to point out that compared with ChIP-seq experiments, CUT&Tag experiments usually have much less DNA (in particular DNA with size between 150-300 bp) immunoprecipitated by control IgGs. We agree with the Reviewer that the controls need to be sequenced for ChIP-seq experiments, but in our view, it is unnecessary for CUT&Tag experiments.

Fig. R1. Electrophoresis analyses of DNA immunoprecipitated by CUT&Tag experiments after PCR amplification but before purification and size selection. Note that the Spt6 results are not included in this manuscript.

- Why is Tox4 cKO displaying still a TSS enrichment? Tox4 deletion is expected to be complete, isn't it?

Gene deletion using the Cre-LoxP system is efficient but can never be 100%. The efficiency of deletion is determined by the activity of the cell type- or tissue-specific promoter that drives Cre expression and the location of the inserted loxP sites. The efficiency of Tox4 deletion in the current study is decent (Fig. 1b), and consequently, Tox4 occupancy in KO cells markedly reduced compared to that in control cells, (Fig. 5a, b and S6b). The residual Tox4 signal upon *Tox4* deletion may be contributed by cells without efficient *Tox4* deletion, residual Tox4 in *Tox4* KO cells, and non-specific pulldown by the TOX4 antibody.

Fig. 1b. Western blot comparing cellular level of Tox4, total, Ser-5 phosphorylated and Ser-2 phosphorylated Pol II in control and Tox4 cKO thymocytes. Left: representative pictures of Western blot; Right: a bar graph comparing relative level of total Pol II, Pol II (Ser-2p), Pol II (Ser-5p) and Tox4 in control and cKO cells quantified by ImageJ. Data shown are the mean \pm S.D. of 3 independent experiments. Statistical significance was determined with a two-sided Student's t-test. NS: $P \geq 0.05$, * $P < 0.05$, ** $P < 0.01$ and *** $P < 0.001$.

Fig. 5a, b. **a** Genome-wide meta-gene profiles and heatmaps of CUT&Tag comparing chromatin occupancy of Tox4 in Tox4 cKO versus control (Ctrl) cells. Genes with and without significant Tox4 binding are presented separately. **b** Normalized read distribution of CUT&Tag of Tox4 and Pol II within the *Pknox1*, *Csrp1*, *Rabif* loci in Tox4 cKO versus control cells.

Fig. S6b. Genome-wide meta-gene profiles and heatmaps of CUT&Tag comparing chromatin occupancy of Tox4 in cKO versus control cells.

- Previous studies (Austenaa et al 2015, Cortazar et al 2019, Cossa et al 2020) identified/suggested the presence of the PTW phosphatase also at the TES, where it is relevant for termination. Why is TOX4 not detected there by CUT&Tag?

TOX4 usually occupies genes from TSSs to several kilobases downstream of TESs. In addition, TOX4 occupancy on TSSs usually is higher than that on gene bodies and TESs. Therefore, it is sometimes hard to tell if Tox4 occupies the TESs of some genes by judging from the snapshots because of the high occupancy near the TSSs. With that being said, very low Tox4 occupancy on the TESs of some genes may simply reflect a fact or due to low TOX4 occupancy on those whole genes. To show Tox4 occupancy on TESs, we have added 2 more snapshots (*Csrp1* and *Rabif*) to Fig. 5b in the revised manuscript.

Fig. 5b. Normalized read distribution of CUT&Tag of Tox4 and Pol II within the *Pknox1*, *Csrp1*, *Rabif* loci in Tox4 cKO versus control cells.

3. RNAPII CUT&Tag experiments technically raise similar questions.

- A metagene plot for all genes including Ctrl, Tox4 cKO and a negative control plot should be shown.

As responded above, isotype matched IgG control was included for each group of our CUT&Tag experiments but not sequenced (Fig. R1). In addition, we have also generated metagene plots of all genes for all the CUT&Tag experiments, and placed them in the corresponding supplementary Figures in the revised manuscript.

- Additionally, the Tox4-bound/nonbound stratification appears particularly misleading for this specific RNAPII binding analysis. For example, it shows that Tox4-nonbound genes have no RNAPII, suggesting that Tox4 promotes RNAPII binding, which doesn't appear to be the case when looking at the Tox4 cKO metagene. The authors should solve this apparent discrepancy. We appreciate the Reviewer's comment, have generated metagene plots of all genes for all the CUT&Tag experiments, and placed them in the corresponding supplementary Figures in the revised manuscript.

- Along the same line, is Tox4 binding stratifying to RNAPII-bound genes? Or to expressed genes? Proper correlation analyses should be performed.

We found that TOX4 is capable of directly binding Pol II, although some other factors (including the PAF1 complex) are capable of facilitating its binding (unpublished data). Therefore, it is possible that Tox4 binding stratifies to RNAPII-bound genes.

- Data supporting the claim that Tox4 loss induces loss of RNAPII occupancy are very weak, both in the metagene and in the shown genome browser tracks. I think the actual relevant phenotype is the increase of pSer2-RNAPII along the gene. Still, this is probably indirect, as the proposed target of the PTW phosphatase is pSer5-RNAPII. For this reason, I think the authors should perform also pSer5-RNAPII CUT&Tag.

We agree with the Reviewer that the effect of Tox4 loss on total Pol II occupancy is not great, which is likely due to inactive transcription of the majority of the DP cells (DP rearrangement and DP selection) and the dual roles of TOX4 in transcription.

We have also performed CUT&Tag experiments for Ser-5 phosphorylated (Ser-5p) Pol II in control and Tox4 KO DP cells by following the Reviewer's advice and found a slight increase of Pol II (Ser-5p) occupancy in KO cells relative to that in control cells (Fig. 6g, h, i, j and S8f).

Fig. 6g, h, i, j. **g** Genome-wide meta-gene profiles and heatmaps of CUT&Tag comparing chromatin occupancy of Pol II (Ser-5p) in cKO versus control cells. **h**, **i** Volcano plots comparing Pol II (Ser-2p) occupancy changes near TSSs (**h**) and TESs (**i**) of Tox4-bound genes in cKO versus control cells. **j** Normalized read distribution of CUT&Tag of Tox4, total Pol II, Pol II (Ser-2p) and Pol II (Ser-5p) within the *Rmnd5a*, *Cd8a*, *Dapk3* and *Cdkn2d* loci in cKO versus control cells.

Fig. S8f. Genome-wide meta-gene profiles and heatmaps of CUT&Tag comparing chromatin occupancy of Ser-5 phosphorylated Pol II in cKO versus control cells.

With respect to if Ser-5 or Ser-2 is the bone fide target of the PTW/PP1 complex, we found by in vitro phosphatase assay with purified proteins that TOX4 facilitates dephosphorylation of Pol II CTD serines 2 and 5 by PP1 phosphatases (Liu et al. Communications Biology, 2022). In addition, TOX4 or Tox4 loss showed comparable effects on cellular level of Pol II (Ser-2p) and that of Pol II (Ser-5p) in K562 cells (Liu et al. Communications Biology, 2022) or murine DP T cells (Fig. 6k and I). Together, these results suggest that Pol II CTD Ser-2 is likely to be one of the bona fide targets in cells.

Fig. 6k and I. **k, I** Individual comparison of free (**k**) and chromatin-bound (**I**) Pol II in control and cKO cells by Western blot. Left: representative pictures of Western blot; Right: a bar graph comparing relative level of total Pol II, Pol II (Ser-2p), Pol II (Ser-5p) and Tox4 in control and cKO cells quantified by ImageJ. Data shown are the mean \pm S.D. of 3 independent experiments. Statistical significance was determined with a two-sided Student's t-test. NS: $P \geq 0.05$, * $P < 0.05$, ** $P < 0.01$ and *** $P < 0.001$.

We are aware that some people have proposed that Pol II CTD Ser-5 is a direct target of PTW/PP1 complex. (1) Lee et al. showed that PTW/PP1 complex is capable of dephosphorylating Pol II CTD Ser-5 in vitro (Figure 5) (Lee et al., JBC, 2010); the authors did not analyze the effect on Pol II CTD Ser-2 simply because they used Ser-5p as an example to

support the conclusion of one of their earlier studies that PP1 phosphatases are capable of dephosphorylating both Ser-5p and Ser-2p (Washington et al., JBC, 2002). (2) Ciurciu et al. showed that PNUTS knockout in *Drosophila* increased Pol II CTD Ser-5p (Figure 7B), but the authors did not analyze Ser-2p on the assumption that PNUTS1-PP1 prefers Ser-5p over Ser-2p (Ciurciu et al., PLOS Genetics, 2013) because it was reported in an earlier study that WDR82 binds Ser-5p (Lee et al., Mol Cell Biol, 2008). (3) Landsverk et al. showed clear increase of both Ser-5p and Ser-2p upon WDR82 knockdown (Figure 3E), although the increase of the latter is smaller compared to that of Ser-5p (Landsverk et al., Cell Reports, 2020). In summary, there is no data against Pol II CTD Ser-2p as a direct target of PTW/PP1, and some of the previous studies have ignored analyzing effect of PTW/PP1 on Ser-2 phosphorylation for unjustified reasons.

4. The authors mention several times “transcription reinitiation” (line 314 and elsewhere), but this term doesn’t recall any standard description of RNAPII transcription behavior. Do they mean transcription elongation after pause-release (=“late elongation”)? Or RNAPII eviction from pause site (=“premature termination”) and RNAPII reloading at TSS? The concept should be extensively explained in the introduction or where relevant and, more importantly, well supported by data.

We appreciate the Reviewer’s comment. In our manuscript, the process of Pol II dephosphorylation after the completion of a transcription cycle and the reloading at TSSs with initiation factors is referred to as transcriptional reinitiation. We have rewritten related parts by following the Reviewer’s advice.

5. Concerning the extragenic transcription analysis.

- Why are the outputs of RNA-Seq and TT-Seq so different? This could happen, but I find it extremely surprising that the most striking effects are identified by RNA-Seq and not TT-Seq (which should better identify unstable transcripts, as these extragenic transcripts usually are). We appreciate the Reviewer’s comment, and are aware that TT-seq experiments usually identify much more changes than RNA-seq experiments do in cell lines. We find the results using primary T cells a little bit surprising but not extremely surprising. Primary cells in the immune system are known to have lower RNA content compared with other types of primary cells (Jaehning et al., Cell, 1975 & Yamawaki et al., BMC Genomics, et al. 2021). In addition, DPre and DPsel cells, which make up 60.0% and 22.2% of total DP cells, respectively, are not proliferating and likely to be transcriptionally inactive. After only identified 159 differentially expressed gene by RNA-seq (Fig. 5c), we wanted to see if we would be able to identify more changes using TT-seq. It was surprising to us in the beginning to find out that TT-seq did not identify significantly more changes than RNA-seq did, and that the overlap between them was not great. However, the results can be easily explained by low RNA content and inactive transcription of the majority of the DP cells.

- Why are the TT-Seq results commented in the “take-home” sentence of the paragraph and not the more striking RNA-Seq ones?

We’d like to thank the Reviewer for pointing this out. We only mentioned the TT-seq results at that time because TT-seq experiments measure nascent RNA (transcriptional output) while RNA-seq experiments measure stable RNA. We have modified this part in the revised manuscript to justify why TT-seq results other than RNA-seq results were commented.

- Why are only the up-regulated transcripts analyzed?

Actually, both upregulated and downregulated transcripts upon *Tox4* deletion were shown in FigureS7a and b. In each panel, there are 2 groups of stacked bars, and each group contains two stacked bars labeled as *Tox4* cKO and control, respectively. Transcripts upregulated in

Tox4 cKO were included in the stacked bar labeled as Tox4 cKO, while transcripts upregulated in control (downregulated in Tox4 cKO) were included in the stacked bar labeled as control.

Fig. S7a and b. **a** A stacked bar graph showing annotation of significantly up-regulated Tox4 bound (left) and Tox4 nonbound (right) extragenic transcripts in cKO and control cells, respectively, identified by CUT&Tag and TTchem-seq. **b** A stacked bar graph showing annotation of significantly up-regulated Tox4 bound (left) and Tox4 nonbound (right) extragenic transcripts in cKO and control cells, respectively, identified by CUT&Tag and RNA-seq.

- Some gene browser track examples here would help visualizing the phenotype. We have added 4 tracks to Fig. S7c and d by following the Reviewer's advice.

Fig. S7c and d. **c** Normalized read distribution of TTchem-seq within two extragenic loci directly regulated by Tox4 in cKO versus control cells. **d** Normalized read distribution of RNA-seq within two extragenic loci directly regulated by Tox4 in cKO versus control cells.

6. The authors repeatedly refer to a revised manuscript to back or comment their findings.

These referrals are impossible to evaluate, so the authors should either remove them or describe better the "K562 phenotype" they refer to. On the other side, most of the mechanistic analyses appear to be a confirmation of that revised/unpublished manuscript, which somehow diminishes the actual novelty of the findings reported here.

At the time of the submission of this manuscript, our manuscript with respect to the role of TOX4 in transcriptional regulation in K562 cells was under revision. The paper had been published when we started the revision of this manuscript. In the revised version of the manuscript, we either cited the paper (Liu et al., Communications Biology, 2022) or described the results as findings in K562 cells.

With respect to if the consistency between the results of K562 cells and the results of murine DP cells, described in the current manuscript, diminishes the actual novelty of the findings in the current manuscript, we disagree with the Reviewer. Cell lines are known to be different from primary cells in size (Cultured cancer cells are much larger.) and chromatin accessibility, and large cells need to produce and maintain higher amounts of RNA and protein to sustain biomass and function although the genome content often remains constant (Samuel Marguerat and Jürg Bähler, Trends Genet, 2012). For example, Jurkat cells (a human ALL cell line) are ~5 times larger than primary human T cells, and MEL cells (a murine erythroleukemia cell line) are at least 5 times larger than primary murine erythroid cells. Therefore, in our view, results from cell lines need to be validated by results from primary cells. In addition, there are clear differences between murine and human cells in mechanisms of gene regulation, for example, the mechanisms of X chromosome inactivation, so it is also important to know if the role of Tox4 in gene regulation is evolutionarily conserved between mouse and human. Moreover, the role of TOX4 in development is unknown, and this manuscript also provides insights into the role of TOX4 in T cell development.

7. Generally speaking, why is a key regulator of RNAPII transcription (=binding over 7,500 genes) mediating the up-/down-regulation of only very few of them? I believe this could be in part due to inherent secondary compensation effects upon TOX4 deletion. To investigate this hypothesis, at least some of the TOX4-dependent transcription events should be assessed upon shorter-term depletion methods.

We appreciate the Reviewer's comment, and agree with the Reviewer that secondary compensation effects of TOX4 deletion on Pol II transcription cannot be ruled out. We previously have tried acute TOX4 depletion using the auxin-inducible degron (AID) system, but the resulting AID-TOX4 expressing cell line cannot be used because of leaky degradation (Liu et al. Communications Biology, 2022). Nevertheless, the consistency among the results from the TOX4 KO cell line, the AID-TOX4 cell line and Tox4 cKO murine DP T cells suggest not only that they are reliable but also that the effects of TOX4 on Pol II transcription may be direct.

Minor points.

- Multiple studies about the PP1-PNUTS-TOX4-WDR82 complex (including the original Lee et al 2010) refer to it as the "PTW phosphatase" (or PP1/PTW) complex. I would recommend the authors to use this term, instead of the "PP1C" abbreviation, which usually refers to PP1 catalytic subunits.

We agree with the Reviewer, and have changed PP1 complex to PTW/PP1 complex by following the Reviewer's advice.

- The authors should clarify that all scatter plots in Figures 1 and 2 are representative experiments, otherwise the accompanying bar charts would look discrepant.

We agree with the Reviewer, and have clarified in Figure Legend by following the Reviewer's advice.

- The authors mention Figures "S3a and S3b" which don't exist (line 176). Similarly, also Figure "S7a-d" and "S8a-d" are miscalled (lines 299, 332).

We'd like to thank the Reviewer for pointing the mistakes out, and have corrected them in the revised manuscript.

- What is the exact take-home message of Figure 3e? The authors comment it as a proof of "no effect of global transcription output", but what it mostly shows is that there's relevant variability between replicates. How were the experiments/replicates normalized?

We appreciate the Reviewer's comment, and would like to thank the Reviewer for pointing this out. There is variability between the two replicates, but the difference between control and TOX4 cKO within each replicate is small. With the inactive transcription of the majority of the DP cells (e.g., DPre and DPsel) also being considered, in our view, it is fair to state "no effect of global transcription output".

With respect to the normalization of the TT-seq data, we normalized the data by sequencing depth. We have done TT-seq for more than 20 times, and consider ourselves very experienced with this technique. Yeast RNA were added for normalization by cell number. However, most of the time (or 90% of the time), they introduce more biases other than reducing them for unknown reason. I have communicated with Dr. Jesper Svejstrup, one of the experts of TT-seq, about this, but people in the Svejstrup group are also having the same problem (see Fig. R2). Therefore, we only use yeast RNA for normalization when the trend with and without normalization are the same. In the case of Tox4, the variations among the samples became greater when normalized using yeast RNA (although the trends are the same), so we decided not to use the results.

From: Ming Yu <mingyu@sjtu.edu.cn>
Subject: Fwd: Questions about spike-in used in TTchem-seq
Date: October 14, 2021 at 17:16
To: aiwei aiwei.wu@sjtu.edu.cn

Begin forwarded message:

From: Jesper Qualmann Svejstrup <jsvejstrup@sund.ku.dk>
Subject: Re: Questions about spike-in used in TTchem-seq
Date: October 14, 2021 at 4:23:55 PM GMT+8
To: Ming Yu <mingyu@sjtu.edu.cn>

Dear Ming,
I agree with you that the spike-ins are not always helpful. They are essential if one wants to claim a global change (up or down) of all genes, but if it is the pattern of transcription that changes, it is often not helpful to use 'normalised' data.
I hope this helps,
Jesper

Jesper Q Svejstrup FRS
University of Copenhagen

From: Ming Yu <mingyu@sjtu.edu.cn>
Sent: Thursday, October 14, 2021 10:16:36 AM
To: jsvejstrup@sund.ku.dk <jsvejstrup@sund.ku.dk>
Subject: Questions about spike-in used in TTchem-seq

[You don't often get email from mingyu@sjtu.edu.cn. Learn why this is important at <http://aka.ms/LearnAboutSenderIdentification>.]

Dear Dr. Svejstrup,

I hope that everything is going on well with you.

This is Ming Yu, a principal investigator at Shanghai Jiao Tong University, China. We recently did some TTchem-seq experiments using yeast total RNA as spike-in, and are troubled by the spike-in because it frequently brought bias into the experiments. We feel that spike-in in each sample may not have equal chance getting sequenced. Would you please let me know your thought on this?

Thank you very much.

Best,

Ming

Fig. R2. Emails between Dr. Svejstrup and I regarding the normalization of TT-seq.

- For clarity, the legend of Figure 3h could rather label RNASeq and TTSeq as "(mature) mRNA level" and "nascent mRNA level", respectively.

We'd like to thank the Reviewer for the suggestion, and have modified the labels by following the Reviewer's advice.

- What are Tox4-bound genes (and/or the ones up-/down-regulated upon Tox cKO) doing? A GO Term analysis should be attempted.

We have tried GO analysis using the Tox4 direct targets identified by CUT&Tag and RNA-seq, but found no significant enrichment. In the revised manuscript, we have also tried GO analysis using the Tox4 direct targets identified by CUT&Tag and TT-seq, and significantly enriched pathways are displayed in Fig. 5i and j.

Fig. 5i and j. Pathway analysis results of upregulated (i) and downregulated (j) genes upon Tox4 deletion.

- I think the single-cell analysis part could be moved before the Tox4 CUT&Tag part. This would help the overall flow of the manuscript.

- The decrease of Cdk1, Ccnb1, Ccnb2 in the DPblast population doesn't appear very striking. Is there some supporting statistics here? This should also be validated by alternative means (e.g., qPCR).

We appreciate the Reviewer's comment. In the revised manuscript, we have moved the part related to scRNA-seq (Fig. 4) ahead of the part related to Tox4 CUT&Tag (Fig. 5). In addition, we have purified DPblast (Cd4⁺CD8⁺CD69^{lo}FSC^{hi}) cells, extracted RNA and performed qRT-PCR. The results are consistent with the scRNA-seq results (Fig. S5a-g). DPblast cells are rare, making up ~0.32% of DP cells (Fig. S5a). To obtain enough cells for qRT-PCR, we had to lower the threshold for sorting CD69⁺ cells. If a higher threshold was used, the results will be more striking. Moreover, there is a positive correlation between G2M score and Cdk1 expression (Fig. S5h). Together, these results suggest that low Cdk1 activity may contribute to the proliferation defect of DPblast and lead to cell cycle exit.

Fig. S5a and b. **a** Representative FACS plots of sorting DP blast cells (CD4+CD8+FSChiCD69lo) from control and cKO mice. **b-g** Comparison of mRNA level of Cd8a (**b**), Cd8b1 (**c**), Cdk1 (**d**), Ccnb1 (**e**), Ccnb2 (**f**) and Ccna2 (**g**) by quantitative RT-PCR. Data shown are the mean \pm S.D. of 4 independent experiments. Statistical significance was determined with a two-sided Student's t-test. $P < 0.05$, $**P < 0.01$ and $***P < 0.001$.

Fig. S5h. A scatter plot showing a positive correlation between calculated G2M score and Cdk1 expression. Pearson correlation coefficient is placed on top of the plot. Dots are colored according to the predicted cell cycle phase.

- The passage about the overexpression of ribosomal protein genes (line 278) is speculation with scarce relevance for the rest of the story, so it should be probably omitted.

We'd like to thank the Reviewer for the suggestion, and have deleted this part by following the Reviewer's advice.

Reviewer #2 (Remarks to the Author):

In this study, Liu and colleagues report the role of Tox4, a regulatory subunit of protein phosphatase 1 (PP1), in T cell development in mice by controlling the expression of the genes involved in the T cell receptor (TCR) signaling pathway. From their mouse genetic and functional genomics data, the authors also suggest that Tox4 regulates transcriptional elongation and reinitiation.

The experiments have been designed elegantly and thoughtfully. The language is lucid, and overall, the paper is written quite well. However, some major and minor concerns need to be addressed before publication.

We greatly appreciate the Reviewer's comments. We have performed all the required experiments to address the Reviewer's concerns and revised the manuscript by following the Reviewer's advice. Additionally, we have changed the title of the manuscript to emphasize its relevance to Developmental Biology.

Major Comments:

1. Authors suggest that the loss of Tox4 impairs T cell development in part by downregulating TCR signaling genes, e.g., Cdk8a. Then based on their genomics data—the genome-wide distribution of Pol II, Spt5, Spt5-pThr806—the authors conclude that Tox4 functions as a suppressor of elongation. How can the authors reconcile these two seemingly contradictory conclusions?

We appreciate the Reviewer's comment. We have found that TOX4 facilitates Pol II CTD serines 2 and 5 and SPT5 Threonine 806 (Thr-806) dephosphorylation, so that it is capable of

positive and negatively affects transcription on the same gene and/or different genes. Specifically, dephosphorylation of Pol II CTD serine 2 and SPT Thr806 decreases early elongation rate, while dephosphorylation of Pol II CTD serines 2 and 5 facilitates Pol II recycling and reinitiation (Liu et al., Communications Biology, 2022). In addition, we also found that TOX4 also stimulates elongation through unclear mechanisms (Liu et al., Communications Biology, 2022). Consequently, TOX4 loss mainly decreases expression of highly expressed genes, which are dependent on the high frequency of reinitiation. Moreover, TOX4 loss mainly increases expression of genes with low expression level, which are the least dependent on the frequency of reinitiation but are the most sensitive to elongation rate change compared with genes with high and medium expression level (Fig5. k-n) (Liu et al., Communications Biology, 2022). *Cd8a* is one of the highly expressed genes in DP blast cells, so that Tox4 KO decreases its expression.

2. Fig 1b – There is a difference between the increase in Pol II CTD-pSer2 and -pSer5 upon depletion of Tox4. The increase is more pronounced in Pol II-pSer2—maybe due to the low level of pSer2 in control cells—despite the equal protein loading (as observed by Actin). An explanation for the differences would be beneficial for the reader.

We appreciate the Reviewer's comment. We have repeated the experiments 2 more times, and found that the increases of the cellular level of p-Ser2 and p-Ser5 are actually comparable (Fig. 1b, 6k and l). It was reported previously that CTD Ser-5 is a substrate of the PTW/PP1 complex (Ciurciu et al., PLOS Genetics, 2013). We found by in vitro phosphatase assay with purified proteins that TOX4 facilitates dephosphorylation of Pol II CTD serines 2 and 5 by PP1 phosphatases (Liu et al. Communications Biology, 2022). In addition, TOX4 or Tox4 loss showed comparable effects on cellular level of Pol II (Ser-2p) and that of Pol II (Ser-5p) in K562 cells (Liu et al. Communications Biology, 2022) or murine DP T cells (Fig. 6k and l). Together, these results suggest that Pol II CTD Ser-2 is also likely to be one of the bona fide targets of PTW/PP1 in cells.

Fig. 1b. Western blot comparing cellular level of Tox4, total, Ser-5 phosphorylated and Ser-2 phosphorylated Pol II in control and Tox4 cKO thymocytes. Left: representative pictures of Western blot; Right: a bar graph comparing relative level of total Pol II, Pol II (Ser-2p), Pol II (Ser-5p) and Tox4 in control and cKO cells quantified by ImageJ. Data shown are the mean \pm S.D. of 3 independent experiments. Statistical significance was determined with a two-sided Student's t-test. NS: $P \geq 0.05$, * $P < 0.05$, ** $P < 0.01$ and *** $P < 0.001$.

Fig. 6k and I. **k, I** Individual comparison of free (**k**) and chromatin-bound (**I**) Pol II in control and cKO cells by Western blot. Left: representative pictures of Western blot; Right: a bar graph comparing relative level of total Pol II, Pol II (Ser-2p), Pol II (Ser-5p) and Tox4 in control and cKO cells quantified by ImageJ. Data shown are the mean \pm S.D. of 3 independent experiments. Statistical significance was determined with a two-sided Student's t-test. NS: $P \geq 0.05$, * $P < 0.05$, ** $P < 0.01$ and *** $P < 0.001$.

3. Undoubtedly, it is well accepted that the Pol II CTD Ser2 phosphorylation accumulates beyond the polyadenylation site (PAS, authors defined as TES). The pSer2 distribution by CUT&Tag in Fig 5d and 5g resembles Pol II and/or Pol II CTD pSer5 distribution—peaks around the TSS with almost no accumulation of the same beyond the PAS. Therefore, it is recommended to check the anti-pSer2 antibody. Moreover, the observed non-canonical distribution of pSer2 puts all of the data in Fig 5 related to pSer2 in question. This reviewer suggests redoing the experiments to identify the canonical distribution of pSer2 in control and Tox4 KO cells and then analyzing accordingly.

We understand the Reviewer's concern. Actually, we had used the most recognized Pol II (Ser-2p) antibodies, 3E10, for the CUT&Tag experiments in DP cells. The reason why Pol II (Ser-2p) signal downstream of TESs is no higher than that around TSSs, which is different from what people have normally seen (mainly in cell lines), is twofold: (1) we used CUT&Tag instead of ChIP-seq, and (2) we used murine primary T cells other than cultured cancer cells, which have greatly increased volume and global chromatin decompaction so that data quality of primary cells usually is lower than that of cell lines. Moreover, we have compared Pol II (Ser-2p) CUT&Tag and ChIP-seq side-by-side using K562 cells before, and found that CUT&Tag is better than ChIP-seq (Liu et al., Communications Biology, 2022 & Figure R3 below).

Actually, we have compared CUT&Tag and ChIP-seq experiments dozens of times, and found that CUT&Tag is always much better. We had done close to 1000 ChIP-seq experiments, and have become uncomfortable with this technique after learning that formaldehyde fixation disassociates proteins from chromatin in 2017 during the Mechanisms of Eukaryotic Transcription Meeting at Cold Spring Harbor. We have done close to 1000 CUT&Tag experiments since 2020, and have great confidence in this technique over ChIP-seq.

Fig. R3. Comparison of ChIP-seq and CUT&Tag of Pol II (Ser-2p) in K562 cells. (a) Metagene profiles of ChIP-seq and CUT&Tag of Pol II (Ser-2p) in K562 cells, (b) to (h) Normalized reads distribution of ChIP-seq and CUT&Tag of Pol II (Ser-2p) within *ZVTB48*, *KLHL21*, *PHF13*, *THAP3*, *DNAJC11*, *EEF1A1*, *ODC1*, *ACTB*, *UBP1*, *PAICS* and *PFKFB4* loci.

4. How does the author conclude that Tox4 may also facilitate transcriptional reinitiation by measuring Pol II and Pol II pSer2 (though it is not right) occupancy on chromatin? The level of non-chromatin-bound hypophosphorylated Pol II critically controls initiation. Therefore, the authors should measure the level of non-chromatin bound (free) pSer2 and/or pSer5 in both cytoplasmic and nuclear fractions in control and Tox4 KO conditions.

We appreciate the Reviewer's comment, and have performed the Western blot experiments by following the Reviewer's advice. Comparison of level of free and chromatin-bound Pol II individually in control and cKO cells by Western blot discovered increased level of both free and chromatin-bound Pol II (Ser-2p) and Pol II (Ser-5p), while level of both free and chromatin-bound total Pol II was unaffected (Fig. 6k and l). These results are in agreement with results of the Western blot using whole cell lysate (Fig. 1b) and results of the CUT&Tag experiments of Pol II (Ser-2p) and Pol II (Ser-5p) (Fig. 6d, g, j, S8e and f), but different from what we have obtained using K562 cells showing increased level of free total Pol II but decreased level of chromatin-bound total Pol II (Liu et al., Communications Biology, 2022). However, results of the total Pol II CUT&Tag experiments showed decreased occupancy of it upon *Tox4* deletion (Fig. 6a and S8d). Therefore, the difference can be easily explained by the small number of genes showing significant total Pol II occupancy changes upon *Tox4* deletion (Fig. 6b and c). These results suggest that similar to what TOX4 does in K562 cells, Tox4 may also restrict elongation and facilitate reinitiation in murine DP cells.

Fig. 6d, g, j. **d, g** Genome-wide meta-gene profiles and heatmaps of CUT&Tag comparing chromatin occupancy of Pol II (Ser-2p) (**d**) and Pol II (Ser-5p) (**g**) in cKO versus control cells. **j** Normalized read distribution of CUT&Tag of Tox4, total Pol II, Pol II (Ser-2p) and Pol II (Ser-5p) within the *Rmnd5a*, *Cd8a*, *Dapk3* and *Cdkn2d* loci in cKO versus control cells.

Fig. S8e and f. **e, f** Genome-wide meta-gene profiles and heatmaps of CUT&Tag comparing chromatin occupancy of Ser-2 phosphorylated (**e**) and Ser-5 phosphorylated (**f**) Pol II in cKO versus control cells.

5. While the authors explain the decrease and increase in Pol II, Spt5, and Spt5-pThr806 occupancy near the TESs, does it upstream, around, or beyond the TESs (PASs)? Moreover, it would be beneficial for the readers if the authors could mark the TES in the schematic diagrams of the example genes in the browser tracks.

We appreciate the Reviewer's comment. Total Pol II occupancy clearly decreased upstream and downstream of TESs (Fig. 6a and S8d), while that of Pol II (Ser-2p) and Pol II (Ser-5p) clearly increased (Fig. 6b, c, S8e and f). The changes of Spt5 and p-Spt5 Thr806 around the TESs were minimal although more genes showed significantly increased p-Spt5 Thr806 occupancy than those showed decreased p-Spt5 Thr806 occupancy (Fig. 7g). We have also marked the location of the TESs for all the gene tracks in the revised manuscript.

Minor Comments:

1. Some minor typographical errors and spelling mistakes need to be rectified. For example, on page number 8, "To facilitates" would be "To facilitate".

We'd like to thank the Reviewer for pointing it out, and have corrected it in the revised manuscript.

2. Authors can consider rephrasing and/or shortening some sentences. That way, readers would be more invested while reading. On the other hand, bigger sentences might distract the readers.

We'd like to thank the Reviewer for pointing it out, and have made some changes accordingly in the revised manuscript.

3. Fig 1d. and 1f. – Both the "% of DP cells" and "cell number of DP cells" are increasing

compared to the DN cells. The authors probably refer to the decrease in the conditional KO DP cells in both the figures compared to the control cells. It would be beneficial if the authors rephrase or reconstruct the sentence describing Fig 1c, d, and f.

We'd like to thank the Reviewer for pointing it out, and have made the changes in the revised manuscript.

Reviewer #3 (Remarks to the Author):

While the role of TOX and TOX2 HMGbox proteins in development and functioning of the immune system has been the focus of much work, nothing has been reported in regard to the in vivo role of TOX4 in lymphocyte development. Using a conditional knockout model, this manuscript describes a very comprehensive and well performed analysis of a role for TOX4 in the thymus, in part based on somewhat similar results that have been obtained in previous work on human lymphoblast CML K562 cells.

Overall, the work is of high quality, has appropriate statistical analyses, uses state-of-the-art available technologies, and the findings novel. The biggest limitation in terms of significance is the relatively modest effect of loss of TOX4. This is somewhat surprising when compared to loss of other TOX family members in certain immune (and other) contexts, where loss of these nuclear proteins can have profound effects on cell differentiation/lineage commitment. The authors repeatedly state that one reason for this is the heterogeneity of double positive thymocytes. It is not at all clear to this reviewer what heterogeneity is being referred to and how this explains the results. This should be clarified. And it is all the more surprising given the profound enhancement of Pol II ser-2P on bulk cells.

We appreciate the Reviewer's comment, and have clarified the related part in the revised Manuscript. The modest effect of *Tox4* loss on T cells development may be either a fact or due to the inefficient *Tox4* deletion by *Lck-Cre* (Figure 1b). We stated that DP cells are heterogenous because they include at least three populations, DP blast (DPbla), DP rearrangement (DPre) and DP selection (DPsel). Mechanistically, TOX4 is capable of negatively and positively impact transcription by facilitating Pol II CTD serines 2 and 5 dephosphorylation (Liu et al., Communications Biology, 2022). Consequently, genes with high expression level are more dependent on TOX4, whereas genes with medium expression level may not be sensitive to TOX4 loss despite of significant increase of Pol II (Ser-2p) (Fig. 5k-n). Moreover, DPre and DPsel, which make up the majority of the DP cells, are likely to be transactionally inactive (Jaehning et al., Cell, 1975 & Yamawaki et al, BMC Genomics, et al. 2021). Therefore, the effect of *Tox4* deficiency on T cell development is modest.

Figure 1b. Western blot comparing cellular level of Tox4, total, Ser-5 phosphorylated and Ser-2 phosphorylated Pol II in control and Tox4 cKO thymocytes. Left: representative pictures of Western blot; Right: a bar graph comparing relative level of total Pol II, Pol II (Ser-2p), Pol II (Ser-5p) and Tox4 in control and cKO cells quantified by ImageJ. Data shown are the mean \pm S.D. of 3 independent experiments. Statistical significance was determined with a two-sided Student's t-test. NS: $P \geq 0.05$, * $P < 0.05$, ** $P < 0.01$ and *** $P < 0.001$.

In this regard is it also possible that TOX4 plays a modest accessory function in the thymus but has a more profound effect on mature T cells? Do the KO T cells in these mice proliferate normally in response to TCR stimulation (Fig. 2 only shows steady state LN T cells)? It is also stated that TOX4 unlike other family members likely regulates transcription in an HMG-box-independent manner because of limited observed chromatin accessibility changes. The logic here is not clear nor supported by data. As the HMG-box is the DNA-binding domain one would expect that it likely plays some role in targeting the protein (along with presumably distinct protein binding partners) to the various CUT&TAG sites identified. And given the strong conservation of this sequence it seems likely it also is key to TOX4 function, regardless of the ultimate mechanism of action in regulating transcription.

We'd like to thank the Reviewer for the great suggestion. We have monitored expression of activation markers, i.e., CD69 and CD25, and performed CFSE assay to determine if mature T cells respond normally to TCR stimulation upon Tox4 deletion. We found that activation and proliferation of CD4 cells was unaffected (Fig. 3a-e) while those of CD8 cells were slightly impaired (Fig. 3f-j). In the revised manuscript, we have included those results in the new Fig. 3.

Fig. 3a-e. **a, b** Representative plots of flow cytometric analyses of CD69 (**a**) and CD25 (**b**) expression of stimulated CD4⁺ lymphocytes from control and cKO mice. **c, d** Bar graphs comparing frequency of CD69 (**c**) and CD25 (**d**) positive CD4⁺ lymphocytes from control and cKO mice. **e** Left: Representative plots of flow cytometric analyses of CFSE labeled CD4⁺ lymphocytes from control and cKO mice. Right: A bar graph comparing frequency of CFSE labeled CD4⁺ lymphocytes from control and cKO mice. Statistical significance was determined with a two-sided Student's t-test. NS: $P \geq 0.05$.

Fig. 3f-j. **f, g** Representative plots of flow cytometric analyses of CD69 (**f**) and CD25 (**g**) expression of stimulated CD8+ lymphocytes from control and cKO mice. **h, i** Bar graphs comparing frequency of CD69 (**h**) and CD25 (**i**) positive CD8+ lymphocytes from control and cKO mice. **j** Left: Representative plots of flow cytometric analyses of CFSE labeled CD8+ lymphocytes from control and cKO mice. Right: A bar graph comparing frequency of CFSE labeled CD8+ lymphocytes from control and cKO mice. Statistical significance was determined with a two-sided Student's t-test. NS: $P \geq 0.05$, * $P < 0.05$ and ** $P < 0.01$.

With respect to the functional dependency of Tox4 on its HMG-box, we agree with the Reviewer that we need to be more cautious when making a statement. We have found that the interaction between TOX4 and PP1 phosphatases is dependent on its C-terminus but not the HMG-box (Liu et al. Communications Biology, 2022) and that Tox4 loss is unlikely to affect transcription through modulating chromatin accessibility. The more accurate statement (compared with our previous statement) should be that Tox4 is unlikely to regulate transcription through modulating chromatin accessibility.

Other issues:

Figure 1b has a band in the TOX4 blot. Is this TOX4 or a nonspecific band? If the latter this needs to be documented. Of course, reduced efficiency of TOX4 deletion could also explain the relatively modest effect. Can the KO make an in-frame truncated protein (would it be detected by the antibody)?

The band in Figure 1b is very likely to be the remaining Tox4 from cells without *Tox4* deletion. We have also done Western blot using membrane corresponding to whole lanes of an SDS-PAGE gel, and found no truncated Tox4 expression after *Tox4* deletion (Fig. R4).

Fig. R4. Comparison of Tox4 expression in control and cKO mice by Western blot.

Controls throughout are fl/fl mice. Just the most basic finding in terms of % and numbers of DPs should be presented for control cre+ mice to eliminate potential artifacts caused by overexpression of cre (which has been noted for some strains).

We'd like to thank the Reviewer for the suggestion. We actually had also compared Tox4^{fl/fl}, Tox4^{fl/+};Lck-Cre and Tox4^{fl/fl};Lck-Cre side by side, and found that Cre overexpression is unlikely to affect the phenotype (Fig. R5).

Fig. R5. Flow cytometric analysis of expression of CD4, CD8, CD25 and CD44 in thymocytes.

Minor: It is stated that the cre mice are lck distal promoter Is that a typo as lck proximal promoter expression is usually earlier?

We'd like to thank the Reviewer for pointing it out, and have changed "distal" to "proximal".

Reviewers' comments:

Reviewer #1 (Remarks to the Author):

I acknowledge that the authors addressed almost all the raised issues by either discussing and defending their work or by performing a number of crucial experiments. I support the publication of the manuscript, although some points still deserve comments.

1. The new experiments/mice solidified the previous data about the decreased cellularity, while not shedding new light about its mechanism (i.e., apoptosis). I acknowledge the way the relevant paragraphs have been rephrased.
2. I understand the authors' point concerning negative controls for CUT&Tag (and I find convincing the included agarose gels showing no enrichment), in contrast to traditional ChIP-Seq controls. Still I believe that sequencing a negative control would provide a valuable baseline for the metagenes/heatmaps. For example, the gene body enrichment of Tox4 in Fig. 5a is around 0.1 rpm. Would this mean that Tox4 is enriched in gene bodies or not? Is this a signal or a background?
3. I appreciated the authors' discussion about the Ser5 vs. Ser2 targeting by PTW/PP1. I encourage the authors to incorporate at least part of it in the discussion.
4. All issues have been addressed.
5. Concerning the labeling of Fig. S7a-b, I still believe this is a bit misleading. Why not calling the conditions "upregulated vs. downregulated" upon Tox4 cKO instead of "upregulated in Tox4 cKO vs. upregulated in control". The authors use "upregulated vs. downregulated" also in other passages, so I would uniform this.
6. Concerning the novelty of this story, I agree that this manuscript sheds new light on Tox4 role in development. Still, the mechanistic part is grossly superimposable to the K562 manuscript, if I'm not mistaken, so the novelty of this part is lower.
7. I would incorporate part of the authors' response to the manuscript discussion.

All minor points have been properly addressed.

Reviewer #2 (Remarks to the Author):

The authors have addressed all of my concerns. The manuscript can be accepted in its current format.

Reviewer #3 (Remarks to the Author):

The authors have made significant and substantive changes in response to the reviewer critiques, including new experimentation. That being said, the major weakness of the study remains the fact that despite extensive and the excellent analyses of these knockout mice, some of the biological effects are relatively modest. Whether this is due to compensatory mechanisms, the efficiency of deletion, or simply that TOX4 plays an accessory but not obligatory role in the studied biological contexts is not clear. Taken together, the data do support that TOX4 functions in the cellular proliferation rather than differentiation associated with the DN to DP thymocyte transition, potentially as a regulator of Cdk1. TOX4 is most unlike other family members in sequence and expression pattern. Consistent with that, the data makes a compelling case that unlike other TOX family members where studied, the major mechanism of action of TOX4 is not regulating chromatin accessibility.

After the DP stage, there are no changes in the frequency of SPs but a reduction in numbers as might be expected by reduced DP, suggesting no significant effect on positive selection, although there is some increase in the CD4:CD8 ratio in the thymus and LN, in part due to a CD8 survival defect, of unknown cause. However, in terms of the latter it appears that the investigators are not gating on post-selection CD8SP (i.e., CD3hi), and thus the CD4:CD8 ratio likely includes pre-DP CD8ISP and may not be accurate; this also makes the Fig. 2 thymus analysis problematic. Giving the changes in the DN to DP transition, looking by FACS at CD8 ISP should be included.

It is also worth noting that the authors make two general arguments in their rebuttal for the modest effects they report- either deletion of TOX4 is inefficient in their mouse model (although many have used this cre strain to reveal gene function, Fig. 1b does show incomplete deletion of TOX4 in total thymocytes), or there is little effect on DP cells as they are more transcriptionally silent and include different developmental subpopulations. The former could be tested by either using cre reporter strains or at a minimum quantitatively assessing the extent of cre-mediated DNA deletion at various stages of T cell development to see if there is clear selection against undeleted cells subsequent to the DP stage.

Unless the investigators can show that the reduced expression of Cd8 genes at the DP stage is reflected at the protein level- not at all obvious from their FACS staining- the argument that that affects the function of the cells or is responsible for the subsequent altered ratio of CD4:CD8 T cells does not follow. It is also not clear how reduced Cd8 gene expression would lead to increased apoptosis of CD8 T cells (line 262). And concluding that Tox4 regulates the expression of TCR signaling because of this reduction in Cd8 gene expression is unsupported.

In summary, a comprehensive and well executed study, but in the end an incremental advance that identifies a new player in one aspect of T cell development and adds to an understanding of how TOX4 can regulate gene expression.

Reviewers' comments:

Reviewer #1 (Remarks to the Author):

I acknowledge that the authors addressed almost all the raised issues by either discussing and defending their work or by performing a number of crucial experiments. I support the publication of the manuscript, although some points still deserve comments.

We'd like to thank the Reviewer for the comments, and have performed additional experiments to address the Reviewer's concerns. Please find our point-by-point response below.

1. The new experiments/mice solidified the previous data about the decreased cellularity, while not shedding new light about its mechanism (i.e., apoptosis). I acknowledge the way the relevant paragraphs have been rephrased.

2. I understand the authors' point concerning negative controls for CUT&Tag (and I find convincing the included agarose gels showing no enrichment), in contrast to traditional ChIP-Seq controls. Still I believe that sequencing a negative control would provide a valuable baseline for the metagenes/heatmaps. For example, the gene body enrichment of Tox4 in Fig. 5a is around 0.1 rpm. Would this mean that Tox4 is enriched in gene bodies or not? Is this a signal or a background?

We understand the Reviewer's concern. If the deficiency of a factor would have a major effect on chromatin accessibility, it would be necessary to perform CUT&Tag experiments in control and cKO cells using isotype matched IgG to obtain baselines for the downstream analysis. However, we found in this study that Tox4 deficiency had little effect on chromatin accessibility (Fig. S11).

With respect to why we have never performed CUT&Tag experiments using control IgG in control and cKO cells, respectively, and sequenced the immunoprecipitated DNA, the reason is the very low quantity of DNA immunoprecipitated by the CUT&Tag experiments using control IgG. We previously have found that for human cancer cell lines, CUT&Tag experiments using control IgG (mouse, human or rat) always produce very low signal (barely visible on agarose gel) (Figure R1). Therefore, even when comparing control cells and knockdown (KD) or knockout (KO) cells by CUT&Tag, we always only perform 1 control experiment each time using IgG and control cells simply to make sure that background of the experiments is still low. However, we had never sequenced the control samples before. We have also found that background of experiments using control IgG and murine primary cells is higher than that of experiments using control IgG and human cancer cell lines, but is still much lower than that of experiments using non-control antibodies and murine primary cells (Figure R2A). Therefore, we have always performed 1 control experiment each time using IgG and control cells when comparing control and Tox4 cKO cells by CUT&Tag.

Fig. R1. Electrophoresis analyses of DNA immunoprecipitated by CUT&Tag experiments using human K562 cells after PCR amplification but before purification and size selection.

To address the Reviewer's concerns, we have sequenced all the 6 control samples of our CUT&Tag experiments by following the Reviewer's advice. As stated above, the control experiments were only performed using control cells (Tox4^{fl/fl}) at that time, and mouse, rabbit and rat IgG were used in 2 experiments, respectively (Figure R2A). We found that three of them (2 using mouse IgG and 1 using rabbit IgG) showed relative high signal near TSSs, which is still only ~1/10 of that of experiments using non-control antibodies, while the rest of them showed very low signal (2 using rat IgG, and 1 using rabbit IgG) (Figure R2B and C). We were able to draw at least two major conclusions from the results: (1) signal of CUT&Tag experiments using rat or rabbit IgG generally is very low and the signal variation between experiments using IgG from the same species is simply random, and (2) signal of control experiments using mouse IgG are relatively high in mouse cells, but the signal variation between experiments is still random. To minimize random variation, all our CUT&Tag experiments have two biological replicates, and reproducibility between the biological replicates is great with most of the R values close to or greater than 0.9 (Fig. S6a, S9a-c and S10a and b). Importantly, only high confident peaks obtained from the two biological replicates were used for the generation of metagene plots and heatmaps, and only occupancy changes greater than 20% are considered significant.

In summary, experiments using isotype matched IgG in control and cKO cells to obtain baselines for the downstream analysis are unnecessary for the current study. Additionally, judging from the plot of the cKO, the gene body enrichment of Tox4 in the control of Fig. 5a is weak signal.

Fig. R2. CUT&Tag experiments of control IgG using DP from control and cKO mice, respectively are dispensable for the current study. A. Electrophoresis analyses of DNA immunoprecipitated by CUT&Tag experiments after PCR amplification but before purification and size selection. Note that the Spt6 results are not included in this manuscript. B. Genome-wide meta-gene

profiles of 6 CUT&Tag experiments of the current study using control IgG and DP cells from control mice. C. Genome-wide meta-gene profiles of 4 CUT&Tag experiments of Pol II using DP from control and cKO mice, respectively, and 2 CUT&Tag experiments of mouse IgG using DP cells from control mice.

3. I appreciated the authors' discussion about the Ser5 vs. Ser2 targeting by PTW/PP1. I encourage the authors to incorporate at least part of it in the discussion.

We have incorporated a part of our response to the Reviewer's comment to the Discussion part of the revised manuscript by following the Reviewer's advice.

4. All issues have been addressed.

5. Concerning the labeling of Fig. S7a-b, I still believe this is a bit misleading. Why not calling the conditions "upregulated vs. downregulated" upon Tox4 cKO instead of "upregulated in Tox4 cKO vs. upregulated in control". The authors use "upregulated vs. downregulated" also in other passages, so I would uniform this.

We'd like to thank the Reviewer for pointing this out. We had made those Figures by following two previously published papers (Austena et al., Mol Cell, 2015 & Nat Struct Mol Biol, 2021), and still cannot figure out a better way to present the data. Therefore, we have modified the legend of Fig. S8a and b (the original Fig. S7a and b) to improve the readability of the revised manuscript.

6. Concerning the novelty of this story, I agree that this manuscript sheds new light on Tox4 role in development. Still, the mechanistic part is grossly superimposable to the K562 manuscript, if I'm not mistaken, so the novelty of this part is lower.

We'd like to thank the Reviewer for the comments, and are aware that the Reviewer's view is widely supported by researchers in the field of gene regulation. However, we believe that data from primary cells are more valuable than those from immortalized cancer cell lines, which are several times larger than primary cells in size and usually with chromatin abnormality, and data from immortalized cancer cell lines have to be validated using primary cells.

7. I would incorporate part of the authors' response to the manuscript discussion.

We have incorporated a part of our response to the Reviewer's comment to the Discussion part of the revised manuscript by following the Reviewer's advice.

All minor points have been properly addressed.

Reviewer #2 (Remarks to the Author):

The authors have addressed all of my concerns. The manuscript can be accepted in its current format.

Reviewer #3 (Remarks to the Author):

The authors have made significant and substantive changes in response to the reviewer critiques, including new experimentation. That being said, the major weakness of the study remains the fact that despite extensive and the excellent analyses of these knockout mice, some of the biological effects are relatively modest. Whether this is due to compensatory

mechanisms, the efficiency of deletion, or simply that TOX4 plays an accessory but not obligatory role in the studied biological contexts is not clear. Taken together, the data do support that TOX4 functions in the cellular proliferation rather than differentiation associated with the DN to DP thymocyte transition, potentially as a regulator of Cdk1. TOX4 is most unlike other family members in sequence and expression pattern. Consistent with that, the data makes a compelling case that unlike other TOX family members where studied, the major mechanism of action of TOX4 is not regulating chromatin accessibility.

We'd like to thank the Reviewer for the comments, and have performed additional experiments to address the Reviewer's concerns. Please find our point-by-point response below.

After the DP stage, there are no changes in the frequency of SPs but a reduction in numbers as might be expected by reduced DP, suggesting no significant effect on positive selection, although there is some increase in the CD4:CD8 ratio in the thymus and LN, in part due to a CD8 survival defect, of unknown cause. However, in terms of the latter it appears that the investigators are not gating on post-selection CD8SP (i.e., CD3hi), and thus the CD4:CD8 ratio likely includes pre-DP CD8ISP and may not be accurate; this also makes the Fig. 2 thymus analysis problematic. Giving the changes in the DN to DP transition, looking by FACS at CD8 ISP should be included.

We'd like to thank the Reviewer for pointing these out. In the revised manuscript, we have included CD8⁺ISP (CD8⁺TCR β ⁻) into the analysis by following one of the previously published papers (Gegonne et al., Cell Reports, 2018). We found that Tox4 cKO increased its frequency while minimally affected its number (Fig. 1e-g).

We have also performed additional experiments to analyze proliferation and apoptosis of DN, ISP, DP and SP. The new conclusions are (1) that Tox4 deficiency slightly decreased proliferation of CD8 without impairing that of ISP (Fig. 2a and b), and (2) that Tox4 deficiency minimally affected apoptosis of ISP (Fig. 2e-g). The rest of the conclusions in the revised manuscript remain the same as those in the previous version of the manuscript.

It is also worth noting that the authors make two general arguments in their rebuttal for the modest effects they report- either deletion of TOX4 is inefficient in their mouse model (although many have used this cre strain to reveal gene function, Fig. 1b does show incomplete deletion of TOX4 in total thymocytes), or there is little effect on DP cells as they are more transcriptionally silent and include different developmental subpopulations. The former could be tested by either using cre reporter strains or at a minimum quantitatively assessing the extent of cre-mediated DNA deletion at various stages of T cell development to see if there is clear selection against undeleted cells subsequent to the DP stage.

We'd like to thank the Reviewer for pointing this out. We have sorted DN, ISP, DP and SP thymocytes of control and cKO mice, respectively, and analyzed Tox4 deletion efficiency by Western blot. We found expectedly that Tox4 deletion in ISP, DP and SP is efficient but is inefficient in DN. We have included the results in the revised manuscript (Fig. 1c) and updated the manuscript accordingly.

Unless the investigators can show that the reduced expression of Cd8 genes at the DP stage is reflected at the protein level- not at all obvious from their FACS staining- the argument that that affects the function of the cells or is responsible for the subsequent altered ratio of CD4:CD8 T cells does not follow. It is also not clear how reduced Cd8 gene expression would lead to increased apoptosis of CD8 T cells (line 262). And concluding that Tox4 regulates the expression of TCR signaling because of this reduction in Cd8 gene expression is unsupported. We agree with the Reviewer and would like to thank the Reviewer for pointing this out. To understand how Tox4 deficiency decreases ratio of CD8 to CD4 cells (Fig. 1 n and o) through

impairing proliferation of CD8 thymocytes (Fig. 2a and b) and increasing apoptosis of CD8 cells (Fig. 2e-j), we sorted CD8 thymocytes (TCR β +CD8+) from control and cKO mice, respectively, and performed RNA-seq experiments (n=5). Tox4 deficiency only significantly affected mRNA level of a small subset of genes, and with fold change ≥ 1.5 and FDR < 0.05 , the numbers of downregulated and upregulated genes were 10 and 34, respectively (Fig. S7a and b). Although no pathway was found to be enriched through GO analysis, Cdkn1a (encoding p21), a well-known negative regulator of the cell cycle and a direct target of Tox4 in DP cells (Fig. 5d), was significantly upregulated under Tox4 deficiency, suggesting that p21 upregulation may contribute to the impaired proliferation of CD8 thymocytes. In addition, expression of several apoptosis related genes was significantly affected, including Ntrk3, Gimap7 and Nfia, under Tox4 deficiency, suggesting that dysregulation of these genes may contribute to the increased apoptosis of CD8 cells. We have included the results in the new Fig. S7a and b and updated the manuscript.

In summary, a comprehensive and well executed study, but in the end an incremental advance that identifies a new player in one aspect of T cell development and adds to an understanding of how TOX4 can regulate gene expression.

REVIEWERS' COMMENTS:

Reviewer #1 (Remarks to the Author):

The authors addressed all my concerns. I support the publication of the manuscript.

Reviewer #3 (Remarks to the Author):

The authors have addressed my technical concerns.

REVIEWERS' COMMENTS:

Reviewer #1 (Remarks to the Author):

The authors addressed all my concerns. I support the publication of the manuscript.

Reviewer #3 (Remarks to the Author):

The authors have addressed my technical concerns.

We'd like to thank the Reviewers for their help.